# Understanding Language Prior of LVLMs by Contrasting Chain-of-Embedding

**Lin Long**[*] **Changdae Oh**[*] **Seongheon Park** **Sharon Li**[†]
University of Wisconsin–Madison
{llong,changdae,seongheon_park,sharonli}@cs.wisc.edu
[*]Equal contribution [†]Corresponding author

## Abstract

Large vision-language models (LVLMs) achieve strong performance on multi-modal tasks, yet they often default to their *language prior* (LP)—memorized textual patterns from pre-training while under-utilizing visual evidence. Prior analyses of LP mostly rely on input–output probing, which fails to reveal the internal mechanisms governing when and how vision influences model behavior. To address this gap, we present the first systematic analysis of language prior through the lens of chain-of-embedding, which examines the layer-wise representation dynamics within LVLMs. Our analysis reveals a universal phenomenon: each model exhibits a *Visual Integration Point* (VIP), a critical layer at which visual information begins to meaningfully reshape hidden representations and influence decoding for multimodal reasoning. Building on this observation, we introduce the *Total Visual Integration* (TVI) estimator, which aggregates representational discrepancy beyond the VIP to quantify how strongly visual query influences response generation. Across 60 model–dataset combinations spanning 10 contemporary LVLMs and 6 benchmarks, we demonstrate that VIP consistently emerges, and that TVI reliably predicts the strength of language prior. This offers a principled toolkit for diagnosing and understanding language prior in LVLMs. Code: O

## 1 Introduction

Modern large vision-language models (LVLMs) (OpenAI, 2025; Comanici et al., 2025; Bai et al., 2025; Zhu et al., 2025) have extended the boundaries of AI applications at an unprecedented rate. Their remarkable capability in solving highly complex vision-language tasks originated from the internalized rich unimodal knowledge during the pre-training (Radford et al., 2021; Oquab et al., 2024; Brown et al., 2020) and also from the strong multimodal alignment (Liu et al., 2023; Dai et al., 2023; Zhu et al., 2024). Despite their successes, a central challenge remains: LVLMs are prone to over-relying on their *language prior* (LP)—the statistical patterns memorized during large-scale language model pretraining—while under-utilizing the actual visual evidence (Fu et al., 2024; Lee et al., 2025; Luo et al., 2025). This imbalance often results in hallucinations, shortcut reasoning, and brittle generalization when tasks truly demand visual grounding. For example, when asked "What color is the banana?", an LVLM may confidently answer "yellow" even if the banana in the image is green, demonstrating that the model defaults to its LP. Recent studies (Yin et al., 2024; Liu et al., 2024d; Lee et al., 2025) further show that such LP reliance persists across diverse tasks.

Understanding and quantifying LP in LVLMs is thus critical, both for diagnosing their limitations and for guiding the design of more reliable multimodal systems. However, current approaches to analyzing LP primarily rely on input–output probing. For instance, Lee et al. (2025) and Luo et al. (2025) constructed datasets with counterfactual visual input to measure models' performance under challenging visual grounding scenarios, while Deng et al. (2025) evaluate models on modality-conflicting queries to assess modality preference. While useful, such coarse input-output analyses have fundamental limitation to investigate LP of LVLMs in-depth, because: (1) they ignore the rich latent representations inside the model, which may inform how textual and visual signals are integrated, and (2) they cannot disentangle *where* in the model the LP begins to interfere with effective visual integration, leaving per-sample mechanistic interpretation (Bereska & Gavves, 2024) elusive.

Motivated by this, we propose a new framework for understanding and quantifying language prior, which leverages the chain-of-embedding—the sequence of hidden representations across LVLM

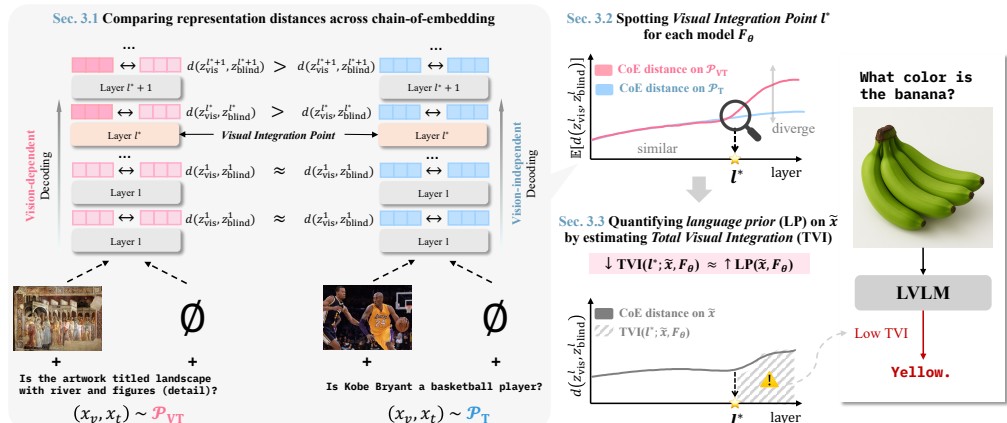

Figure 1: **Framework Overview.** For data from two distributions $\mathcal{P}_{\text{VT}}$ (vision-dependent) and $\mathcal{P}_{\text{t}}$ (vision-independent), we extract chain-of-embedding for two queries w/ and w/o visual input, and use the expected representation distance to spot *visual integration point* $l^*$. Then, estimating *total visual integration* based on $l^*$ allows us to quantify LP of an LVLM per sample.

layers. Making use of these latent representations is essential, because they provide direct insight into the inner mechanisms of LP, beyond surface-level outputs. Specifically, our framework contrasts embeddings from vision-text inputs ($Z_{\text{vis}}^l$) with those from vision-removed inputs ($Z_{\text{blind}}^l$), at each layer $l$. Based on the contrastive chain-of-embedding, we reveal a striking phenomenon: LVLMs exhibit a ***Visual Integration Point* (VIP)**, a layer at which visual information begins to meaningfully influence the LVLM's decoding process. At and beyond VIP, the distance between $Z_{\text{vis}}^l$ and $Z_{\text{blind}}^l$ increases substantially for vision-dependent tasks, signaling that the model has begun to actively integrate visual evidence to solve the task. In contrast, vision-independent tasks show a smaller such shift. Thus, VIP captures a critical point where visual input begins to exert meaningful influence on inference, revealing the extent to which the model relies on vision or falls back on language priors.

Inspired by observations from VIP, we propose quantifying LP through ***Total Visual Integration* (TVI)**, which measures the effective amount of visual integration that affects the answer decoding of LVLM. Specifically, TVI aggregates distance between contrastive embeddings $Z_{\text{vis}}^l$ and $Z_{\text{blind}}^l$ across all post-VIP layers to measure the cumulative strength of visual integration. Intuitively, TVI is inversely related to the magnitude of LP: models with strong reliance on language priors exhibit low TVI, while those that leverage vision more deeply exhibit high TVI. Through extensive experiments covering 10 contemporary LVLMs and 6 datasets (60 settings combined), we show the universality of the existence of VIP, and that TVI can be a reliable indicator of LP. Moreover, we demonstrate that TVI strongly correlates with performance on benchmarks requiring visual reasoning, outperforming other proxies such as visual attention weights or output divergence. Then, we provide a theoretical interpretation of our measure as well as analytic bounds of it for broader use in practice. We illustrate the overall framework in Figure 1, and summarize our contribution as follows:

1. We present a novel framework that contrasts the chain-of-embedding of an LVLM for fine-grained analysis of the visual integration and language prior of LVLMs.

2. Based on this framework, we show that there is a specific layer, VIP, where an LVLM's behavior undergoes a dramatic change, and observe that post-VIP layers' representations play a key role in quantifying the amount of language prior of an LVLM.

3. Across 10 representative LVLMs and 6 datasets, we consistently demonstrate the existence of VIP, show how we can use it to predict the strength of language prior of an LVLM on a certain sample through TVI, and further present theoretical analyses on our framework.

## 2 PROBLEM STATEMENT

**Basic notations.** Let $\mathcal{D} = \{(x_v, x_t)_i\}_{i=1}^N$ denote a dataset of $N$ image-text queries $(x_v, x_t)$, sampled from a population distribution $\mathcal{P}$. Each tuple $(x_v, x_t)$ consists of a visual input $x_v$, and a natural language query $x_t$, expressed in a prompt form. We distinguish $X_v$ from $x_v$ to denote a random variable and its observation (similarly for $X_t$). Then, we define the data structure as follows.

**Definition 2.1.** *We define $\mathcal{P}_{VT}$ as the **vision-dependent** distribution, consisting of examples where resolving the textual query requires access to the associated visual input. In contrast, $\mathcal{P}_T$ is the **vision-independent** distribution, containing examples where the textual query can be answered correctly without visual information (i.e., the text alone suffices). A sample dataset $\mathcal{D}$ is constructed with $\mathcal{D}_{VT}$ and $\mathcal{D}_T$, each containing at least one element from populations $\mathcal{P}_{VT}$ and $\mathcal{P}_T$, respectively:*

$$\mathcal{D} = \{\mathcal{D}_{VT} \cup \mathcal{D}_T : \min(|\mathcal{D}_{VT}|, |\mathcal{D}_T|) \geq 1\}, \tag{1}$$

*where $|\cdot|$ denotes the cardinality of a set. See examples of $\mathcal{D}_{VT}$ and $\mathcal{D}_T$ in Figure 1.*

Meanwhile, we have an LVLM, $F_\theta = f_h \circ f_L \circ f_{L-1} \circ \cdots \circ f_1 \circ f_0$, parameterized with $\theta$. Here, $f_0$ denotes the composition of the visual encoder, modality connector, and text embedding layer; $(f_1, \ldots, f_L)$ corresponds to the $L$ stacked decoder layers of the LLM; and $f_h$ is an output head. The LVLM maps the multimodal input query $(x_v, x_t)$ to a $|\mathcal{V}|-$dimension probability distribution over the vocabulary space $\mathcal{V}$, from which the most likely answer $\hat{y}$ is obtained via the argmax operator.

**Language prior (LP).** An LVLM has a vast amount of knowledge in its parameters obtained during unimodal pretraining and visual instruction tuning of entire model components. Since the pre-training of LLM backbone is far more extensive in quantity and diversity of data, and total computing budget, *LVLMs are prone to over-reliance on memorized statistical textual patterns without integrating visual information during inference.* Given an input $x$ and an LVLM $F_\theta$, we define the model's reliance on statistical textual patterns as the language prior, $\text{LP}(x, F_\theta)$. Note that LP is more like a latent property that lacks a gold-standard measurement. Therefore, previous work typically approximates how robust an LVLM is against LP through its performance on carefully curated datasets. In contrast, we propose a novel approach that (1) does not require any annotations or careful data curation, (2) tries to quantify LP in a more direct manner, which enables flexible and fine-grained, sample-wise diagnosis for LP of LVLMs. Refer to Appendix A for additional context.

**Our position.** Although there have been recent attempts to analyze LP in LVLMs, they primarily focus on evaluating model predictions on curated datasets (Lee et al., 2025; Luo et al., 2025; Vo et al., 2025), without offering a well-defined or generalizable formulation. We argue that such coarse input-output analysis is insufficient: it cannot reveal how LP manifests within the model nor how it can be rigorously quantified. In particular, prior approaches overlook the rich latent information encoded inside LVLM—intermediate representations that inform how visual and textual signals are integrated and how biases emerge. Making use of these latent representations is essential because they provide direct insight into the inner mechanisms of LP, beyond surface-level outputs. With this motivation, we pose the following research question: ***Can we derive a formal framework to understand and quantify the language prior of LVLMs through the lens of their internal states?***

## 3 METHODOLOGY

### 3.1 CHAIN-OF-EMBEDDING AND REPRESENTATION DISTANCE

In contrast to previous approaches that focus on LVLM output (Rahmanzadehgervi et al., 2024; Vo et al., 2025; Lee et al., 2025; Luo et al., 2025), we leverage the ***chain-of-embedding*** for fine-grained analysis of LVLM, which is defined as a sequence of hidden states across layers, *i.e.*, $(Z^1, \cdots, Z^L)$, where $Z^l = f_l(X_v, X_t) \in \mathbb{R}^{d_z}$ [1] denotes the last-token embedding at $l \in \{1, ..., L\}$ as a contextual summary vector[2]. Notably, we contrast embeddings from two different input constructions as below.

$$Z_{\text{vis}}^l := f_l(X_v, X_t) \qquad \text{(embedding from both visual and textual inputs)}$$
$$Z_{\text{blind}}^l := f_l(\varnothing, X_t) \qquad \text{(embedding from textual input only)}$$

Now, given a distance metric $d$, we analyze the difference between these two embeddings per layer by defining an expected ***representation distance*** and its finite-sample estimator,

$$\mathbf{D}_l(\mathcal{P}_\star, F_\theta) := \mathbb{E}_{(X_v, X_t) \sim \mathcal{P}_\star}[d(Z_{\text{vis}}^l, Z_{\text{blind}}^l)], \quad \mathbf{D}_l(\mathcal{D}_\star, F_\theta) := \frac{1}{|\mathcal{D}_\star|} \sum_{(x_v, x_t)_i \in \mathcal{D}_\star} d(z_{\text{vis}}^{l,i}, z_{\text{blind}}^{l,i}), \tag{2}$$

where $\mathcal{D}_\star$ is $\mathcal{D}_{VT}$ or $\mathcal{D}_T$, and $\mathcal{P}_\star$ is $\mathcal{P}_{VT}$ or $\mathcal{P}_T$.

---

[1]Although $Z^l = f_l(...f_2(f_1(X_v, X_t)))$ is more precise, we slightly abuse the notation for clarity.

[2]Such last-token embeddings integrate information from all preceding tokens and are widely used to investigate model's behavior when generating the next token (Jiang et al., 2024; Tian et al., 2024; Li et al., 2025b).

We adopt the cosine distance by default, though other distance functions, including non-metric distances (Deza & Deza, 2009), can also be valid. An ablation study with alternative metrics is provided in Section 4. Intuitively, $Z_{\text{vis}}^l$ should encode distinctive visual semantics that cannot be inferred from text alone, whereas $Z_{\text{blind}}^l$ primarily reflects the model's default linguistic expectations. However, the degree of this discrimination can depend on how visual information contributes differently to different data, and across different layers $l$ of the model. We elaborate on this in the next section.

## 3.2 VISUAL INTEGRATION POINT HYPOTHESIS

Deep neural networks are known to develop hierarchical representations across layers (Chen et al., 2023; Fan et al., 2024; Jin et al., 2025), where each layer has different types and resolutions of information (Joseph & Nanda, 2024; Skean et al., 2025; Artzy & Schwartz, 2024; Jiang et al., 2025). In this paper, we hypothesize that an LVLM has a *Visual Integration Point* (VIP) $l^*$, a critical layer where the model begins to actively leverage visual information to perform task-specific reasoning. Prior to this point, the model primarily engages in general-purpose processing of visual and textual inputs—visual features may be "seen," but not yet "used" to guide inference, and the interactions between modalities remain shallow. This behavioral shift can be reflected in the representation distances: at and beyond VIP, the distance between $Z_{\text{vis}}^l$ and $Z_{\text{blind}}^l$ increases markedly for vision-dependent tasks ($\mathcal{P}_{\text{TV}}$), signaling that the model has started to utilize visual information to solve the task, while vision-independent tasks ($\mathcal{P}_{\text{T}}$) show smaller such shift. Thus, the notion of VIP captures a key behavior transition inside LVLMs. If such a specific point $l^*$ exists, identifying it allows us to localize where the differences between language-prior-dominated and visually grounded inference start to manifest within the model's internal processing. We formalize this hypothesis below.

**Hypothesis 3.1** (**Existence of the visual integration point**). *Given a distance metric $d(\cdot, \cdot) : \mathcal{Z} \times \mathcal{Z} \to \mathbb{R}$, distributions $\mathcal{P}_{TV}$ and $\mathcal{P}_T$ (Eq. 1), and an LVLM $F_\theta$ with $L$ layers which produces a chain-of-embedding $(Z^1, ..., Z^L)$ given input, let $\mathbf{D}_l$ be an expected representation distance defined as Eq. 2. Then, there exists a visual integration point $l^*$ that discerns $\mathbf{D}_l$ between $\mathcal{P}_{VT}$ and $\mathcal{P}_T$, that is,*

$$\exists \, l^* \in \{1, ..., L-1\} \quad s.t. \quad \begin{cases} \mathbf{D}_l(\mathcal{P}_{VT}, F_\theta) - \mathbf{D}_l(\mathcal{P}_T, F_\theta) > \tau, & \forall \, l \geq l^* \\ \mathbf{D}_l(\mathcal{P}_{VT}, F_\theta) - \mathbf{D}_l(\mathcal{P}_T, F_\theta) \approx 0, & \forall \, l < l^* \end{cases}, \tag{3}$$

*where $\tau \in \mathbb{R}^+$ denote a model-dependent constant threshold for each data distribution[3].*

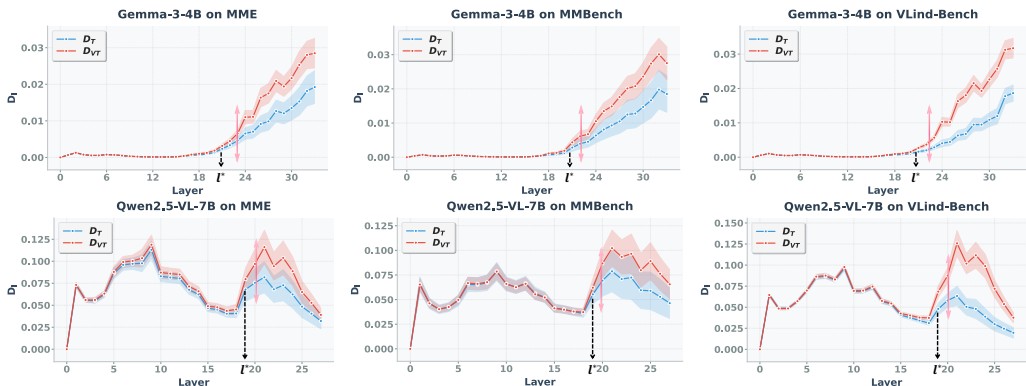

Figure 2: **Visual Integration Point.** We consistently observe that there is a specific layer $l^*$ that clearly distinguish the distance between $Z_{\text{vis}}^l$ and $Z_{\text{blind}}^l$ across two groups $\mathcal{D}_{\text{VT}}$ and $\mathcal{D}_{\text{T}}$.

In Figure 2, we plot the representation distance estimated for two groups: $\mathcal{D}_{\text{VT}}$ (vision-dependent) in red and $\mathcal{D}_{\text{T}}$ (vision-independent) in blue, across all layers in `Qwen2.5-VL-7B` (Bai et al., 2025) and `Gemma-3-4B` (Team et al., 2025). We evaluate on three representative datasets: MME (Yin et al., 2024), MMBench (Liu et al., 2024d), and VLind-Bench (Lee et al., 2025). Since these datasets do not explicitly annotate the degree of visual dependency for each instance ($\mathcal{P}_{\text{VT}}$ *vs.* $\mathcal{P}_{\text{T}}$), we partition each dataset $\mathcal{D}$ into two auxiliary groups: $\mathcal{D}_{\text{VT}} = \{(x_v, x_t) \in \mathcal{D} : F_\theta(x_v, x_t) \neq F_\theta(\varnothing, x_t)\}$ and $\mathcal{D}_{\text{T}} = \{(x_v, x_t) \in \mathcal{D} : F_\theta(x_v, x_t) = F_\theta(\varnothing, x_t)\}$. This split leverages the prediction agreement between multimodal and text-only inputs as a proxy for task type: if two predictions differ, the sample must have demanded visual information to the model, suggesting membership in $\mathcal{P}_{\text{VT}}$ likely.

---

[3]We manually select the $\tau$ and VIP for each model over the observed distances $\mathbf{D}_l$ for analysis convenience (see Appendix B and C for details on this manual selection and an automatic selection method as well).

From Figure 2, we make four key observations: (1) **Existence of VIP**. Representation divergence between $\mathcal{D}_{VT}$ and $\mathcal{D}_{T}$ does not show from the beginning. Instead, for both models, we observe a clear visual integration point ($l^*$), where the representation distance for the $\mathcal{D}_{\text{VT}}$ group rises more sharply compared to $\mathcal{D}_{\text{T}}$ group, marking the onset of genuine multimodal integration; (2) **Behavioral shift across VIP**. We observe a notable increase in the standard deviation of representation distances across VIP. Specifically, before VIP, the model exhibits relatively uniform representation distances across samples, suggesting general-purpose information processing. After VIP, the model's usage of visual information becomes more diverse and instance-dependent to solve a specified task for each query; (3) **VIP is dataset-agnostic**. Within each model, the location of the VIP is relatively consistent across all datasets. For Qwen2.5-VL-7B, the transition consistently occurs around layers 18–20, and for Gemma-3-4B, the transition is around layers 20–22. This stability suggests that the VIP is primarily the LVLM's intrinsic property, not one driven by dataset-specific biases; and (4) **Model-specific patterns**. Despite the shared existence of the VIP, the shape of distance across layers differs across models. In Qwen2.5-VL-7B, representation distance grows relatively smoothly before peaking near the middle-to-late layers and then declines. In contrast, Gemma-3-4B exhibits flat trajectories for many early layers, followed by a steep and monotonic rise after VIP. This suggests that each model has a distinctive hierarchical representation derived from its unique designs.

Overall, these findings highlight not only the universality of the VIP existence, which distinguishes vision-centric decoding (post-$l^*$) from general information-gathering behavior (pre-$l^*$), but also the variability in how different LVLMs distinctively integrate visual information across depth.

### 3.3 Quantifying Language Prior of LVLMs through Total Visual Integration

Although the visual integration point detects the birth of $\text{LP}(x, F_\theta)$, we are also (or even more) interested in how strong $\text{LP}(x, F_\theta)$ is. To quantify this, we define a ***total visual integration*** (**TVI**) estimator in Def. 3.2, which measures the total amount of visual integration that effectively affects the answer decoding of LVLM, and thus is inversely related to LP in nature.

**Definition 3.2** (**Total visual integration estimator**). *For an observed input $x = (x_v, x_t)$, define $x_{vis} := (x_v, x_t)$ and $x_{blind} := (\varnothing, x_t)$. Given an LVLM $F_\theta$ with $L$ decoder layers which produces two sets of chain-of-embedding $(z_{vis}^1, ..., z_{vis}^L)$ and $(z_{blind}^1, ..., z_{blind}^L)$, we define the empirical estimator for the per-sample total visual integration as follows,*

$$TVI(l^*; x, F_\theta) = \frac{1}{L - l^* + 1} \sum_{l=l^*}^{L} \left[ d(z_{vis}^l, z_{blind}^l) \right], \tag{4}$$

*where $z_{vis}^l = f_l(x_{vis})$, $z_{blind}^l = f_l(x_{blind})$, and $d(\cdot, \cdot)$ denotes a distance metric.*

Here $l^*$ marks the VIP layer, where visual information begins to meaningfully influence the model's internal states for visually-grounded decoding. The TVI score then measures the cumulative contribution of visual information by averaging representation distances across all subsequent layers ($l \geq l^*$). The idea behind TVI is that once the model passes the VIP, its internal representations increasingly reflect effective visual grounding, rather than shallow alignment or language-driven statistical patterns. A higher TVI indicates that visual information is more effectively utilized during the response decoding phase, while a lower TVI suggests that the model is more likely to remain text-dominated even after $l^*$. In this sense, TVI provides a holistic measure of how much the model truly uses vision for actual problem solving: *a strong LP corresponds to weak or shallow visual integration (low TVI), while effective multimodal reasoning corresponds to high TVI.*

To investigate the distinction between pre-$l^*$ and post-$l^*$ phases in visual integration, we analyze Spearman's rank correlation between them and answer correctness on VLind-Bench (Lee et al., 2025), which requires visual reasoning. The results in Table 1 show that correlations are weak and statistically less-significant when TVI is computed over pre-$l^*$ layers. In contrast, the post-$l^*$ aggregation yields remarkable correlations with

Table 1: **Spearman's rank correlation between prediction correctness and TVI aggregated from different layers.**

| Model | pre-$l^*$ | post-$l^*$ |
|---|---|---|
| Qwen2.5-VL-7B | 0.1489 ($p = 0.002$) | **0.7241** ($p < 0.001$) |
| Gemma3-4B | 0.4659 ($p < 0.001$) | **0.7174** ($p < 0.001$) |

the prediction correctness, indicating that only after the VIP, the representation distance becomes strongly associated with task performance, thereby serving as a reliable indicator of effective visual integration. In this paper, we stick with post-$l^*$ aggregation in Definition 3.2.

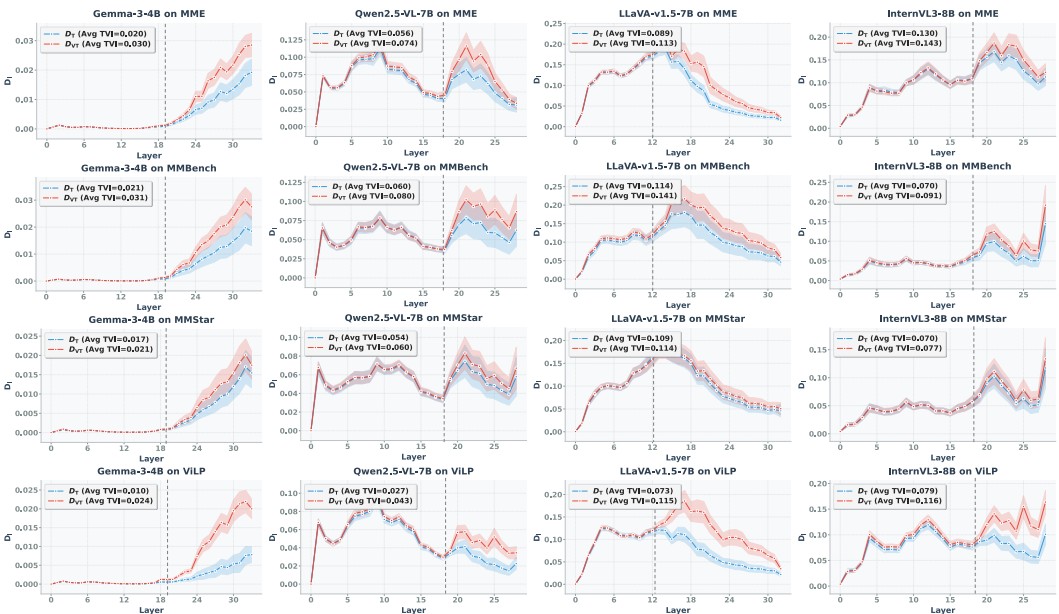

Figure 3: **VIPs of different models observed across different datasets.** Our novel framework, fueled by contrasting chain-of-embedding, allows us to consistently observe VIP across multiple models and datasets, and further enables us to estimate TVI to measure language prior.

Taken together, results from Figure 2 and Table 1 highlight two key insights: (1) the existence of the visual integration point $l^*$, where effective representational shifts starts to happen by integrating visual information, and (2) the strong relationship between post-$l^*$ TVI and downstream performance on vision-dependent tasks. These findings demonstrate that VIP and TVI provide a principled toolkit for analyzing visual integration and language prior in LVLMs. We summarize our findings below.

> **Summary of preliminary findings**
>
> 1. The layer-wise expected representation distance between $\mathcal{D}_{\text{VT}}$ and $\mathcal{D}_{\text{T}}$, *i.e.*, $\mathbf{D}_l(\mathcal{D}_{\text{VT}}, F_\theta) - \mathbf{D}_l(\mathcal{D}_{\text{T}}, F_\theta)$, shows a sudden bump up after a specific layer $l^*$, while marginal before $l^*$.
>
> 2. The aggregated distance $\frac{1}{L-l^*+1} \sum_{l=l^*}^{L} \left[ d(z_{\text{vis}}^l, z_{\text{blind}}^l) \right]$ over post-$l^*$ layers serves as a reliable indicator of language prior, particularly for datasets requiring visual reasoning.

## 4 EXTENDED EXPERIMENTS

Building on the visual integration measurement introduced in the previous section, we conduct additional experiments to assess its empirical validity. Furthermore, we designed a set of in-depth analyses to explore the relationship between visual integration and the language priors in LVLMs.

**VIP consistently emerges across different datasets and models.** We extend the experimental setups described in Section 3 to a broader range of 6 datasets and 10 LVLMs, including `Qwen2.5-VL-7B` (Bai et al., 2025), `InternVL3-8B` (Zhu et al., 2025), `Gemma-3-4B` (Team et al., 2025), `LLaVA-v1.5-7B` (Liu et al., 2024a), `Eagle2.5-8B` (Chen et al., 2025a), `Llama-3.2-11B-Vision`[4], `LLaVA-NeXT-Vicuna-7B` (Liu et al., 2024b), `LLaVA-OV-Qwen2-7B` (Li et al., 2025a) `SmolVLM` (Marafioti et al., 2025), and `InstructBLIP-Vicuna-7B` (Dai et al., 2023). For the datasets, we consider general VQA benchmarks including MME (Chaoyou et al., 2023), MMBench (Liu et al., 2024d), MMStar (Chen et al., 2024), and MMMU (Yue et al., 2024). We also incorporate two benchmarks specifically designed for language prior evaluation, which are VLind-Bench (Lee et al., 2025) and ViLP (Luo et al.,

---

[4]https://huggingface.co/meta-llama/Llama-3.2-11B-Vision-Instruct

2025). This results in a combination of **60 experimental settings**. Implementation details, including data statistics, generation configuration, strategy for VIP selection, etc., are provided in Appendix C. As illustrated in Figure 3, the emergence of VIP is remarkably consistent across all settings: for each model, there exits a clear transition layer $l^*$ where the distance between embeddings $Z_{\mathrm{vis}}^l$ and $Z_{\mathrm{blind}}^l$ increases more significantly for vision-dependent group ($\mathcal{D}_{\mathrm{TV}}$), compared to the vision-independent group ($\mathcal{D}_{\mathrm{T}}$). These results highlight the universality of the VIP existence. Due to the space limit, we defer the complete experimental results to Appendix D.

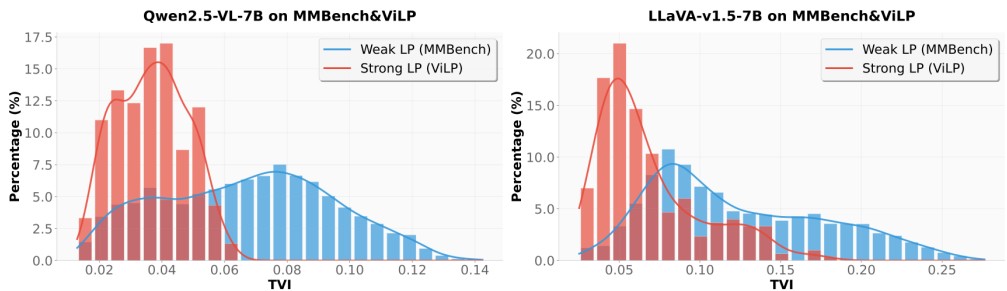

Figure 4: **TVI under language priors of different strengths.** We see that TVI effectively discerns the differences in strength of LP, thereby standing for a reliable measure for LP.

**TVI reliably differentiates strong *vs.* weak language prior.** To examine whether TVI (*c.f.* Definition 3.2) reliably reflects the strength of the language prior, we contrast results on two complementary datasets: ViLP and MMBench. ViLP is intentionally constructed to induce a *strong* LP on the data side by designing queries where plausible answers can often be inferred from textual patterns or statistical correlations without the need for visual grounding. In contrast, MMBench represents a *weak* LP setting, with less misleading questions that encourage stronger visual grounding for task success. As shown in Figure 4, our analysis reveals that datasets with stronger language priors (*e.g.*, ViLP) yield lower TVI values, indicating weaker visual integration in the model, whereas less biased datasets (*e.g.*, MMBench) produce higher TVI values, reflecting stronger use of visual information. This confirms that TVI serves as a reliable quantitative indicator of LP.

**Interventional validation for TVI.** To further verify whether TVI robustly quantifies LP under different inference setups, we conduct a small interventional study. Specifically, we applied an attention-correction-based hallucination mitigation method PAI (Liu et al., 2024c) to `Qwen2.5-VL-7B` as an inference-time intervention, which promotes the model to pay more at-

Table 2: **Downstream performance and TVI before and after intervention.**

|  | Accuracy (%) | TVI |
| --- | --- | --- |
| Before intervention | 50.00 | 0.038 |
| After intervention | 52.33 | 0.144 |

tention to visual features, implicitly increasing visual integration. As shown from the results in Table 2, the intervention not only improves task performance but also yields a substantial increase in TVI. This observation demonstrates that TVI faithfully reflects changes in the model's degree of visual integration, thereby providing robust evidence that it is a reliable metric for quantifying LP.

**Comparison to existing proxy for language prior.** There are alternative approaches to explain LP proposed in previous works, which rely on output-based or attention-based heuristics by assuming (1) LP manifests as high similarity between output tokens generated with and without visual input (Chen et al., 2025b; Xie et al., 2024), or (2) LP arises due to insufficient attention being allocated to visual tokens (Liu et al., 2025). In Table 3, we

Table 3: **Spearman's rank correlation between different metrics and answer prediction correctness.**

| Metric | Qwen2.5-VL-7B | | InternVL-3-8B | |
| --- | --- | --- | --- | --- |
|  | VLind | ViLP | VLind | ViLP |
| TVI | **0.7155** | **0.6335** | **0.6727** | **0.5709** |
|  | ($p < 0.001$) | ($p < 0.001$) | ($p < 0.001$) | ($p < 0.001$) |
| Visual Attention | 0.0871 | -0.0364 | 0.4967 | 0.0746 |
|  | ($p = 0.075$) | ($p = 0.530$) | ($p < 0.001$) | ($p = 0.197$) |
| Output Divergence | 0.2978 | 0.5084 | 0.1627 | 0.5615 |
|  | ($p < 0.001$) | ($p < 0.001$) | ($p < 0.001$) | ($p < 0.001$) |

compare our TVI with two existing approaches (see Appendix C for detailed formulation), average

visual attention and output divergence, by conducting the Spearman's rank correlation analysis between these measures and the correctness of model predictions on two datasets, which all require integrating visual information to produce correct answers. Our TVI consistently exhibits a stronger correlation with output correctness across all datasets and models, suggesting that TVI stands for a reliable indicator of effective visual integration of LVLMs. In contrast, the other approaches show weak and inconsistent correlations in different scenarios.

We argue that both existing approaches fail to directly capture the true impact of visual integration on the model's generation. In the case of visual attention, the model may assign high weights to irrelevant regions of the image rather than the areas required for correct reasoning, and ultimately fall back on its language prior to generate the answer—resulting in inflated attention scores but weak correlation with language prior. Meanwhile, solely measuring output-level discrepancy does not fully capture fine-grained behavior exhibited in internal representation dynamics—differences that are more fundamental in nature than what can be observed from final outputs. It shows the significance of procedural aggregation in TVI. We provide additional visualization analysis in Appendix E.

**Ablations on distance metrics.** To investigate how different choices of distance metric $d$ affect our ability to capture model behavior, we conduct ablation studies with alternative formulations of TVI. As shown in Table 4, TVI remains a strong indicator of model correctness when computed using the L2 distance between latent embeddings. However, when we apply the logit-lens technique (nostalgebraist, 2020)—projecting hidden states at each layer into the output token space and computing divergence between the resulting distributions—the effectiveness of TVI drops significantly. This degradation suggests that such a projection distorts or suppresses the intermediate behavioral differences that occur during decoding. The output space, shaped by the language modeling head, inherently filters latent representations through a decoding-biased lens, which may obscure subtle but meaningful cross-modal integration patterns. These observations reinforce our central

Table 4: **Spearman's rank correlation between correctness and TVI under different distance metrics.** Results are based on evaluations using `Qwen2.5-VL-7B`. All $p$-values are $< 0.001$.

| Metric | VLind | ViLP |
|---|---|---|
| *Embedding-based* | | |
| Cosine Distance | 0.7155 | 0.6335 |
| L2 Distance | 0.7123 | 0.6578 |
| *Output-based (w/ logit-lens)* | | |
| KL Divergence | -0.1693 | 0.2901 |
| JS Divergence | -0.2261 | 0.2942 |

contribution: to faithfully capture the behavioral dynamics of vision-language models, it is essential to examine the internal processing trajectory within the latent representation space, rather than relying on surface-level discrete outputs or their immediate projections. Additional visualization analysis is provided in Appendix E.

**Varying model scales.** We further examine whether our findings generalize across models of different scales. As shown in Figure 5, the VIP consistently emerges across models of varying sizes (4B, 12B, and 27B), underscoring the robustness and generality of our proposed behavioral analysis framework. Interestingly, we also find that the VIP tends to appear at a similar relative depth, which is approximately 60% of the total number of layers, regardless of model size. In addition, after normalizing by the dimensionality of hidden states, we observe that the average normalized TVI is consistently higher in larger models on both $\mathcal{D}_{VT}$ and $\mathcal{D}_{T}$. This suggests

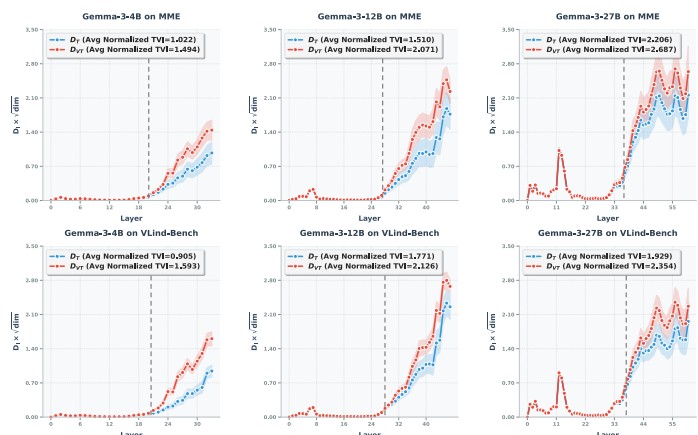

Figure 5: **Varying model scales.** VIP and the dimension-normalized TVI analysis results for three variants of `Gemma-3` model family.

that larger models are more effective at leveraging visual information in a uniform manner across diverse input types, thereby exhibiting greater robustness to misleading language priors. These

observations collectively reinforce the broader applicability of our framework in analyzing visual integration behavior across model scales.

**Practical utility.** We also investigate the practical applicability of TVI here by providing a concrete example to make use of our findings to actually improve LVLMs. Specifically, we leverage TVI as an additional regularization term along with the vanilla cross-entropy loss for next-token prediction during instruction tuning. That is, given the input $x = (x_v, x_t)$ and instruction $y$, we adjusted the original LLaVA training objective (Liu et al., 2023) to:

$$\mathcal{L}(x, y; \theta) = -\log F_\theta(y|x) - \lambda \cdot \text{TVI}(l^*; x, F_\theta), \tag{5}$$

where the strength of the regularization is controlled by $\lambda$ (we set as 0.03). Due to resource limitations, we trained the model on a 60k randomly sampled subset of `llava_v1_5_mix665k`. All other hyperparameters used for visual instruction tuning remain the default of Liu et al. (2023).

The result is shown in Table 5, where the performance improvement indicates that explicitly encouraging stronger visual integration (via TVI) leads to better downstream task performance. This highlights TVI's potential as a helpful training regularizer for improving the visual perception and reasoning of LVLMs in practice.

Table 5: **Effect of TVI regularization on downstream performance, MME dataset (Chaoyou et al., 2023) with LLaVA-v1.5-7B.**

|  | **Perception** | **Reasoning** |
|---|---|---|
| LLaVA-v1.5 | 1369.75 | 298.21 |
| LLaVA-v1.5 w/ TVI | **1400.44** | **321.43** |

**Additional empirical analyses.** In addition, we deepen our understanding of the proposed framework by providing additional analysis in Appendix E, including different aggregation strategies for TVI calculation, instruction-level perturbation, image-text vs. image-only chain comparison, VIP and TVI evolution across training stages, and case studies on TVI failure cases.

## 5 THEORETICAL ANALYSIS

Next, we introduce a new interpretation for our measure, $\mathbf{D}_l(\mathcal{P}_{\text{VT}}, F_\theta) - \mathbf{D}_l(\mathcal{P}_{\text{T}}, F_\theta)$, that locates VIP (Theorem 5.1) and discuss how we can practically employ the expected representation distance (Theorem 5.2) through theoretical analyses. All the proofs and an additional theorem that justifies the use of our empirical representation distance (Lemma F.1) are given in Appendix F.

**Information-theoretic interpretation on representation divergence.** By recasting the representation distance measurement as a density estimation problem, *i.e.*, $d(Z_{\text{vis}}^l, Z_{\text{blind}}^l) \propto -\log \hat{p}_{\text{T}}(Z^l)$ (please see Lemma F.2), we show that the difference in expected representation distances, $\mathbf{D}_l(\mathcal{P}_{\text{VT}}, F_\theta) - \mathbf{D}_l(\mathcal{P}_{\text{T}}, F_\theta)$, which we call representation divergence here, can be interpreted as a relative distributional discrepancy that measures how far the density estimator $\hat{p}_{\text{T}}(Z^l)$, defined by $d(Z_{\text{vis}}^l, Z_{\text{blind}}^l)$, from a population distribution $p_{\text{VT}}(Z^l)$ compared to $p_{\text{T}}(Z^l)$ in Theorem 5.1.

**Theorem 5.1.** *Let $X = (X_v, X_t) \in \mathcal{X}$ be a random variable from $\mathcal{P}_{VT}$ or $\mathcal{P}_T$, and $f_l : \mathcal{X} \to \mathcal{Z}$ be a layer stack from an LVLM $F_\theta$. For $\mathcal{P}_T$, define a density estimator $\hat{p}_T(Z^l) := \mathcal{N}(Z^l; f_l(X_t), I)$, and denote $p_{VT}(Z^l)$ (resp. $p_T(Z^l)$) as the population distribution on $Z^l = f_l(X)$ derived from $\mathcal{P}_{VT}$ (resp. $\mathcal{P}_T$). Then, given $d(Z_1, Z_2) := \frac{1}{2}||Z_1 - Z_2||_2^2$, the difference in the expected representation distances between $\mathcal{P}_{VT}$ and $\mathcal{P}_T$, i.e., $\mathbf{D}_l(\mathcal{P}_{VT}, F_\theta) - \mathbf{D}_l(\mathcal{P}_T, F_\theta)$, can be expressed as follows,*

$$KL(p_{VT}(Z^l)||\hat{p}_T(Z^l)) - KL(p_T(Z^l)||\hat{p}_T(Z^l)) + \bar{\mathbf{H}}, \tag{6}$$

*where $\bar{\mathbf{H}}$ is a constant $H(p_{VT}(Z^l)) - H(p_T(Z^l))$, and $KL(\cdot||\cdot)$ denotes the KL divergence.*

> **Implication.** Theorem 5.1 tells us that $\mathbf{D}_l(\mathcal{P}_{\text{VT}}, F_\theta) - \mathbf{D}_l(\mathcal{P}_{\text{T}}, F_\theta)$ can be interpreted as a relative proximity of the density estimate $\hat{p}_{\text{T}}$ to each distributions $p_{\text{VT}}$ and $p_{\text{T}}$ with an additive constant $\bar{\mathbf{H}}$. Intuitively, the first term, $\text{KL}(p_{\text{VT}}||\hat{p}_{\text{T}})$, can be understood how $\hat{p}_{\text{T}}$ (estimate of $p_{\text{T}}$) far from the true representation distribution on $\mathcal{P}_{\text{VT}}$ while the second term, $\text{KL}(p_{\text{T}}||\hat{p}_{\text{T}})$, is a quality of the density estimation with $\hat{p}_{\text{T}}$ to approximate $p_{\text{T}}$. This expression converts the expected representational distance of the LVLM $F_\theta$ over $p_{\text{VT}}$ and $p_{\text{T}}$ into an information-theoretic divergence, the amount of surprise if we approximate the distribution over $p_{\text{VT}}$ via a blind-representation-centered Gaussian estimator $\hat{p}_{\text{T}}(Z^l) = \mathcal{N}(\cdot; Z_{\text{blind}}^l, I)$.

**Analytic bounds on representation divergence for practical use.** We have assumed a fixed model $F_\theta$ so far. If one is willing to adapt the model to improve its effective visual integration, the analytic bounds in Theorem 5.2 described with $\mathcal{H}$-divergence (see Def. F.4) can be useful.

**Theorem 5.2.** *Let $X = (X_v, X_t) \in \mathcal{X}$ be a random variable of a multimodal input query. Given a stack of LVLM layers $f_l : \mathcal{X} \to \mathcal{Z}$ and a distance metric $d : \mathcal{Z} \times \mathcal{Z} \to [0, 1]$, define a hypothesis $h = d(f_l(X_v, X_t), f_l(X_t)) : \mathcal{X} \to [0, 1]$ and a set of these hypotheses $\mathcal{H}$ that has a pseudo-dimension $c$. Then, for $\mathbf{D}_l(\mathcal{P}_\star, F_\theta) := \mathbb{E}_{X \sim \mathcal{P}_\star}[h(X)]$ with any $\mathcal{P}_{VT}$, $\mathcal{P}_T$, and $\mathcal{P}_M := \frac{\mathcal{P}_{VT} + \mathcal{P}_T}{2}$, and the empirical distributions $\mathcal{D}_{VT} \sim \mathcal{P}_{VT}$ and $\mathcal{D}_T \sim \mathcal{P}_T$ of $N$ samples for each, we have the following bounds w.p. at least $1 - \delta$ for $0 < \delta < 1$,*

$$i) \quad 1 - \mathbf{D}_l(\mathcal{D}_T, F_\theta) - \frac{1}{2}d_{\bar{\mathcal{H}}}(\mathcal{D}_{VT}, \mathcal{D}_T) - \tilde{\mathcal{O}}_\delta \leq \mathbf{D}_l(\mathcal{P}_{VT}, F_\theta), \tag{7}$$

$$ii) \quad \frac{1}{2} - \frac{1}{4}d_{\bar{\mathcal{H}}}(\mathcal{D}_{VT}, \mathcal{D}_T) - \tilde{\mathcal{O}}_\delta \leq \mathbf{D}_l(\mathcal{P}_M, F_\theta) \leq \frac{1}{2} + \frac{1}{4}d_{\bar{\mathcal{H}}}(\mathcal{D}_{VT}, \mathcal{D}_T) + \tilde{\mathcal{O}}_\delta \tag{8}$$

*where $\bar{\mathcal{H}} := \{\mathbb{I}_{|h(X) - h'(X)| > t} : h, h' \in \mathcal{H}, 0 \leq t \leq 1\}$ and $\tilde{\mathcal{O}}_\delta := \mathcal{O}(\sqrt{\frac{1}{N}(\log \frac{1}{\delta} + c \log \frac{N}{c})})$.*

> **Implication.** The first inequality (Ineq. 7) reveals a relationship between two expected representation distances across $\mathcal{P}_{VT}$ and $\mathcal{D}_T \sim \mathcal{P}_T$ with $d_{\bar{\mathcal{H}}}(\mathcal{D}_{VT}, \mathcal{D}_T)$ as a bridge. This tells us that if we want to increase visual integration for an unknown data distribution that require visual reasoning ($\mathcal{P}_{VT}$), we can pursue a greater lower bound of it by decreasing $\mathbf{D}_l(\mathcal{D}_T, F_\theta)$ and $d_{\bar{\mathcal{H}}}(\mathcal{D}_{VT}, \mathcal{D}_T)$ with empirical samples we have. Meanwhile, in a case where we encountered an unknown mixture distribution $\mathcal{P}_M$, the second inequality (Ineq. 8) tells us we can broaden the effective range of $\mathbf{D}_l$ on $\mathcal{P}_M$ by pursuing greater value of $d_{\bar{\mathcal{H}}}(\mathcal{D}_{VT}, \mathcal{D}_T)$.

## 6  CONCLUSION

In this work, we present a formal framework for understanding and quantifying the *language prior* in LVLMs by contrasting the chain-of-embedding between visual and blind contexts. Through this framework, we identify the consistent existence of the ***Visual Integration Point*** (**VIP**), a specific layer at which the model begins to meaningfully incorporate visual context for task-solving beyond the shallow information gathering. Building on this observation, we propose a new metric named ***Total Visual Integration*** (**TVI**), which estimates the degree of effective visual integration and therefore language prior. We conduct comprehensive experiments across 9 LVLMs and 6 datasets, and the results demonstrate that our framework robustly works across models and tasks, providing consistent and interpretable signals about the presence and strength of language prior. Finally, we present some theorems for better understanding and utilization of our framework. We hope that this work sheds light on the internal mechanisms of multimodal models and provides a foundation for diagnosing the language prior, ultimately guiding the development of reliable and responsible LVLMs.

**Limitations.** We set the language prior to LVLMs as our sole target of analysis here, and developed our method based on the representation dynamics of LVLMs. However, there are many other potential biases and vulnerabilities originating from query distribution shifts in the wild, which may induce remarkable changes in the representation space and thus degradation of downstream performance (Verma et al., 2024; Oh et al., 2025a;b; Kim et al., 2025). Reliability of TVI-based language prior estimation should be further validated under realistic distribution shifts.

Besides, our method requires white-box access to the model's internal hidden states and attention patterns. This restricts its applicability to open-weight models and excludes commercial APIs or closed-source systems. However, our framework is primarily designed for model analysis and interpretability research of white-box models, rather than serving as a versatile tool, aiming to shed light on how and when visual information is integrated during inference.

## ACKNOWLEDGEMENT

The authors would like to thank Boyi Li, Haobo Wang, Junbo Zhao, Samuel Yeh, and Jiatong Li for their insightful feedback and helpful discussions throughout the development of this work. Their suggestions greatly contributed to improving the quality of the draft. Changdae Oh, Seongheon Park, and Sharon Li are supported in part by the AFOSR Young Investigator Program under award number FA9550-23-1-0184, National Science Foundation under awards IIS-2237037 and IIS-2331669, Office of Naval Research under grant number N00014-23-1-2643, Schmidt Sciences Foundation, Open Philanthropy, Alfred P. Sloan Fellowship, and gifts from Google and Amazon.

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

# APPENDIX

## CONTENTS

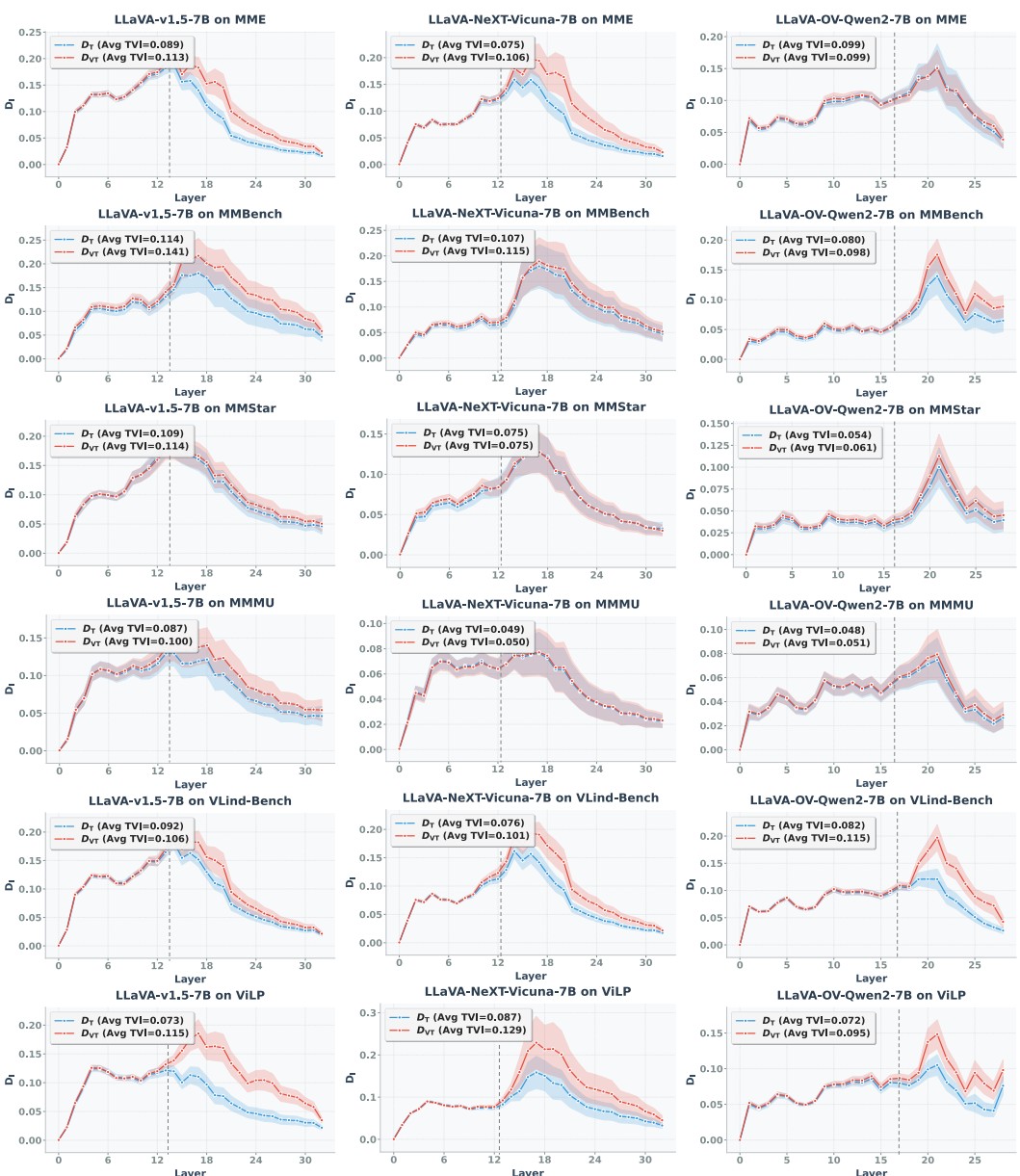

Figure 6: **Complete experimental results.** (Part 1) LLaVA-v1.5-7B, LLaVA-NeXT-Vicuna-7B, and LLaVA-OV-Qwen2-7B on six datasets.

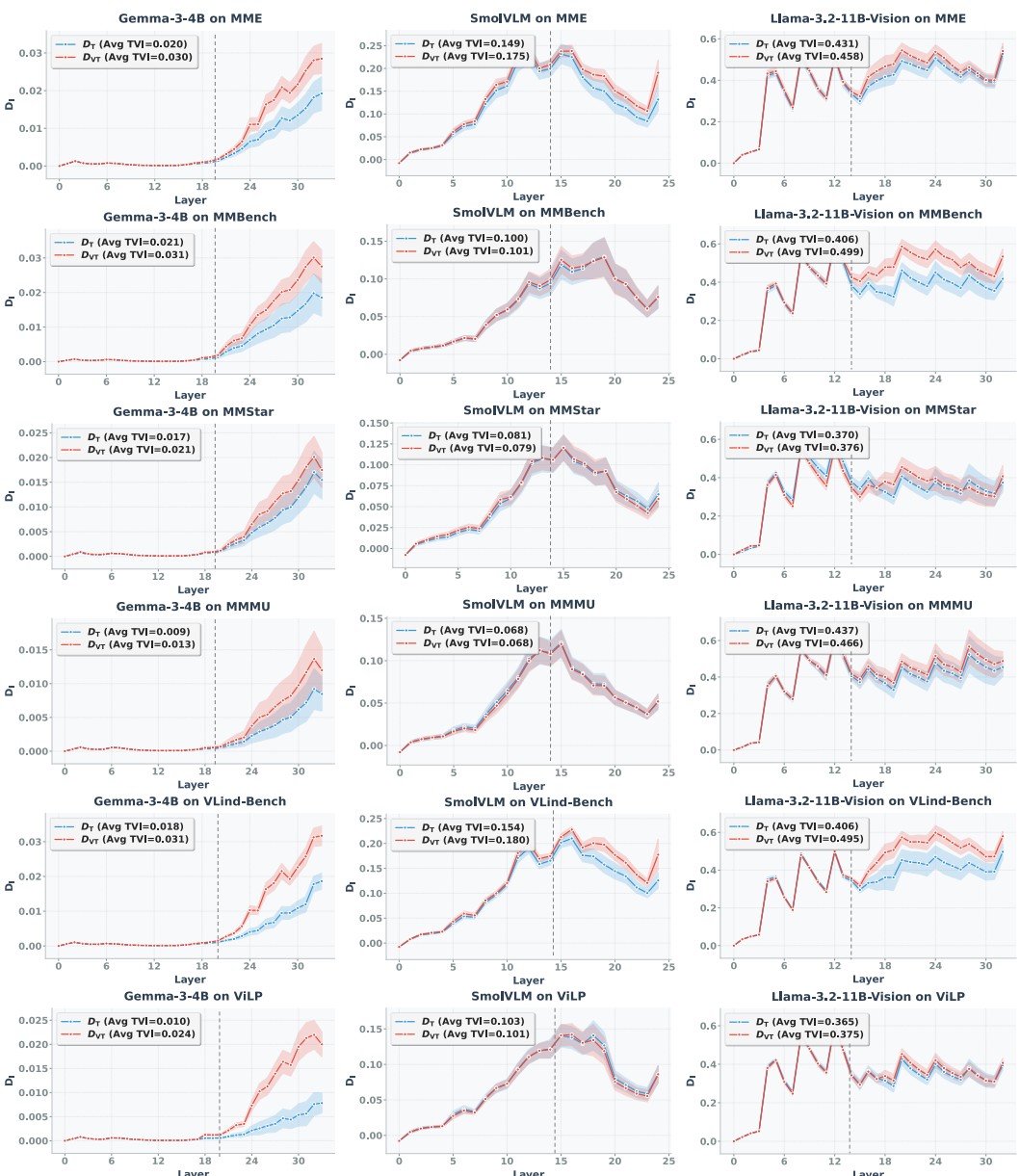

Figure 7: **Complete experimental results.** (Part 2) Gemma-3-4B, SmolVLM, and Llama-3.2-11B-Vision on six datasets.

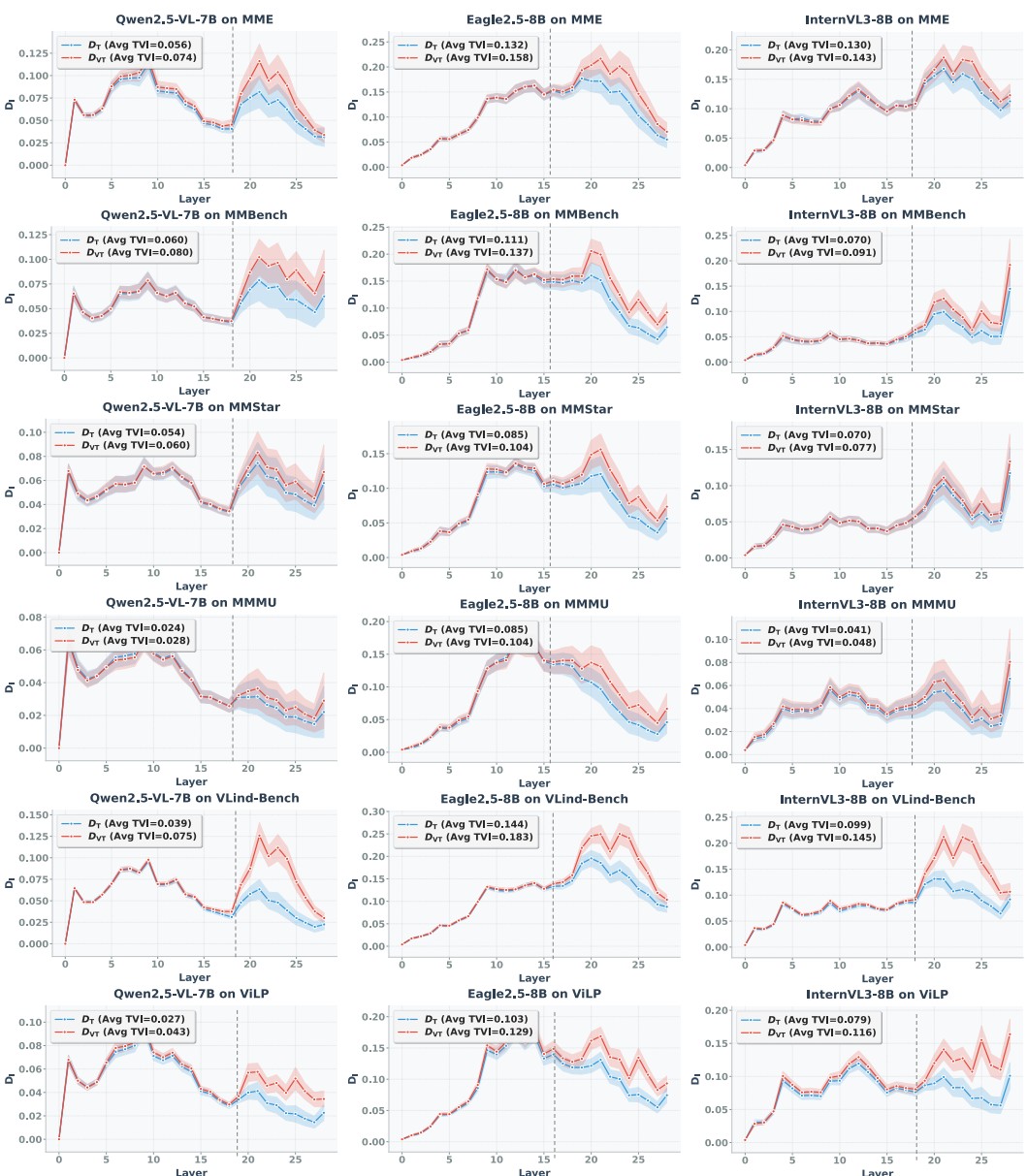

Figure 8: **Complete experimental results.** (Part 3) Qwen2.5-VL-7B, Eagle2.5-8B, and InternVL3-8B on six datasets.

## A   RELATED WORK

**Visual perception in LVLMs.**   Most mainstream LVLMs (Liu et al., 2023; 2024a;b; Bai et al., 2025; Dai et al., 2023) adopt a late-fusion architecture comprising three key components: a vision backbone, a language model that processes both image and text tokens, and a modality adapter that aligns visual representations with the language space. The visual understanding capabilities of these models largely depend on the perception quality of pre-trained vision encoders (*e.g.*, CLIP (Radford et al., 2021), SigLIP (Zhai et al., 2023)) and the reasoning ability of large-scale language models.

Despite promising results on some multimodal benchmarks, this paradigm has been recently challenged due to its poor performance on vision-centric tasks, suggesting that these models often fail to truly see the image (Tong et al., 2024b;a; Vo et al., 2025). A growing body of work has sought to uncover how visual perception operates internally within LVLMs. For example, Bi et al. (2025); Jiang et al. (2025); Neo et al. (2025) examine the layer-wise attention patterns and identify distinct phases of visual integration, typically emerging in the mid-to-late layers. Venhoff et al. (2025) utilize pre-trained sparse autoencoders (SAEs) as analytical tools to show that visual representations gradually align with language representations over depth, converging in the later layers. Complementarily, Fu et al. (2025) apply probing techniques and argue that language decoders in existing LVLMs struggle to effectively leverage the visual features produced by their vision backbones. Their findings suggest that the bottleneck lies not in the availability of visual information but in feature misalignment.

**Language prior in LVLMs.**   One of the most prominent limitations of current LVLMs is their tendency to overly rely on language priors, often generating plausible outputs without grounding in the visual context. This behavior—commonly referred to as the language prior problem—has drawn increasing attention. Recent studies attempt to evaluate this phenomenon by designing datasets that stress-test visual grounding. For example, Lee et al. (2025) and Luo et al. (2025) construct datasets with counterfactual visual inputs to assess whether models can disentangle visual signals from misleading linguistic cues. Similarly, Deng et al. (2025) test modality conflict scenarios to evaluate the model's preference between text and image inputs.

While these works help reveal the presence of language priors, they offer limited insight into the underlying causes. Most current understandings of language prior are based on two widely adopted assumptions: (1) it manifests as high similarity between outputs with and without visual input (Chen et al., 2025b; Xie et al., 2024), and (2) it arises due to insufficient attention allocated to visual tokens (Liu et al., 2025). Building on these assumptions, several works propose methods to mitigate language priors—such as contrastive decoding (Favero et al., 2024) or inference-time attention reallocation (Liu et al., 2024c). Others, like Chen et al. (2025b) and Xie et al. (2024), explore training-time interventions that penalize outputs overly aligned with the model's default language predictions.

## B   DISCUSSION ON VISUAL INTEGRATION POINT

### B.1   NOTE ON VIP DETERMINATION

In Eq. 3, we defined the VIP $l^*$ based on the pre-$l^*$ and post-$l^*$ representation divergences $\mathbf{D}_l(\mathcal{P}_{\mathrm{VT}}, F_\theta) - \mathbf{D}_l(\mathcal{P}_{\mathrm{T}}, F_\theta)$ where the pre-$l^*$ chain-of-embeddings shows nearly zero representation divergence whereas the post-$l^*$ chain-of-embeddings exhibits an effectively large representation divergence defined by positive $\tau$. In practice, we can not access the population distribution $\mathcal{P}_{\mathrm{VT}}$ and $\mathcal{P}_{\mathrm{T}}$, therefore we can not compute the truth expected representation distance $\mathbf{D}_l(\mathcal{P}_\star, F_\theta)$. What we do in practice is estimate that expected representation distance with finite samples $\mathcal{D}_{\mathrm{VT}}$ and $\mathcal{D}_{\mathrm{T}}$, and see the evolution of representation distance gap to manually pick the VIP $l^*$. We observed that this empirical estimator of representation distance (Eq. 2) works well as a measure for determining VIP in general, yet there can be some cases where the fitness of the estimator is bad, e.g., Figure 10. However, even in that case, the point $l^*$, where the empirical representation divergence exceeds a positive constant for all subsequent layers, consistently emerges, and the TVI (Eq. 11) is calculated based on that $l^*$ becomes a strong indicator of LP.

Table 6: **Comparison of VIP detection methods and their effectiveness (Spearman correlations between the resulting TVI and downstream correctness).**

| Model (Layer #) | VIP Detection | MMBench - VIP ($\rho$) | ViLP - VIP ($\rho$) |
|---|---|---|---|
| `Qwen2.5-VL-7B` (28) | Manual | 18 (0.6335) | 18 (0.6335) |
| `Qwen2.5-VL-7B` (28) | Variance-based | 18 (0.6335) | 19 (0.6336) |
| `Gemma-3-4B` (30) | Manual | 20 (0.7970) | 20 (0.7970) |
| `Gemma-3-4B` (30) | Variance-based | 16 (0.7973) | 16 (0.7973) |
| `InternVL3-8B` (32) | Manual | 16 (0.5709) | 16 (0.5709) |
| `InternVL3-8B` (32) | Variance-based | 17 (0.5749) | 20 (0.5949) |

## B.2 ALGORITHMIC ESTIMATION ON VISUAL INTEGRATION POINT

As discussed in §3.2, identifying the VIP provides valuable insights into when visual information begins to influence answer decoding. In most cases, we rely on empirical observation of the representation distance curves to manually determine $l^*$, which proves to be straightforward and interpretable across a wide range of models. However, in situations where manual inspection is impractical — e.g., for large-scale model comparisons or automated pipelines — it may be desirable to estimate the VIP in a data-driven, automatic way. To this end, we introduce an algorithmic rule that can automatically estimate the VIP given a model and a dataset. While not required for our core analysis, this estimation method serves as a practical tool in settings where manual identification of $l^*$ is infeasible.

Below is an estimation strategy based on the test statistic we discussed in Eq 12 of Lemma F.1 by specifying arbitrary significance levels that a user prefers. To be specific, given a pooled sample standard deviation $\hat{\sigma} = \sqrt{\frac{\sigma_{l,\mathrm{T}}^2}{|\mathcal{D}_{\mathrm{T}}|} + \frac{\sigma_{l,\mathrm{VT}}^2}{|\mathcal{D}_{\mathrm{VT}}|}}$, we can define the estimated VIP as follows,

$$\hat{\mathrm{VIP}}(\mathcal{D}, F_\theta) = \arg\min_{l \in \mathcal{L} \setminus L} \frac{\mathbf{D}_l(\mathcal{D}_{\mathrm{VT}}, F_\theta) - \mathbf{D}_l(\mathcal{D}_{\mathrm{T}}, F_\theta)}{\hat{\sigma}_l} \geq \frac{\beta}{l-1} \sum_{k<l} \frac{\mathbf{D}_k(\mathcal{D}_{\mathrm{VT}}, F_\theta) - \mathbf{D}_k(\mathcal{D}_{\mathrm{T}}, F_\theta)}{\hat{\sigma}_k},$$
(9)

where $\sigma_{l,\mathrm{T}}^2 = \frac{\sum_{z \in \mathcal{D}_T} (d(z_{\mathrm{vis}}^l, z_{\mathrm{blind}}^l) - \mathbf{D}_l(\mathcal{D}_{\mathrm{T}}, F_\theta))^2}{|\mathcal{D}_{\mathrm{T}}| - 1}$ is a sample variance on $\mathcal{D}_{\mathrm{T}}$ and $\sigma_{l,\mathrm{VT}}^2$ is similarly defined. Intuitively, the quantity on the left-hand side denotes a deviation-normalized representation distance in the layer $l$, whereas the right-hand side means the historical average of those distances with a weighting coefficient $\beta$, which was set to 2.0 in our case. The above algorithm can be applied to any given dataset, and we conduct comprehensive experiments to show its robust performance across different models and datasets in Table 6 and Figure 9.

From Table 6 and Figure 9, we can see that VIP estimated by the proposed algorithm are quite close to the result of the manually selected one and clearly mark where the two curves start to diverge (based on the divergence plots), and the Spearman correlations between TVIs computed by those estimated VIPs and the downstream prediction correctness are robust across different datasets.

## C IMPLEMENTATION DETAILS

**Models.** We evaluate our framework on 10 publicly available LVLMs, covering a diverse range of architectures and training paradigms. For all models, we use the official instruction-tuned checkpoints available on Hugging Face[5]. To ensure consistent comparison across models, we set the generation temperature to 0.

---

[5] https://huggingface.co/Qwen/Qwen2.5-VL-7B-Instruct
https://huggingface.co/llava-hf/llava-1.5-7b-hf

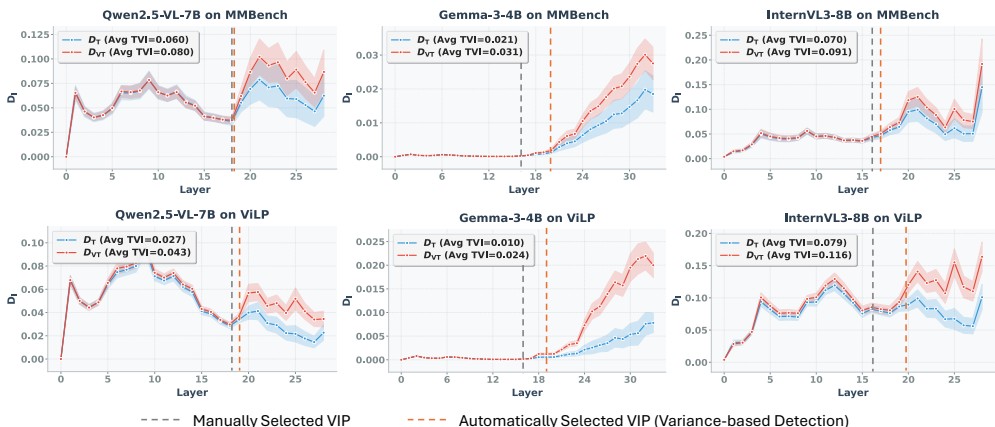

Figure 9: **Comparison between the manually selected VIPs and automatically selected VIPs.**

**Datasets.** Our evaluation spans 6 benchmark datasets, each consisting of either binary ('Yes/No') or multiple-choice questions. This design ensures that the hidden state of the final token—used for representation distance calculations—is closely tied to the model's reasoning process. For MMMU, we use the validation set and filter out samples that involve more than one image or open-ended output to ensure consistency in the evaluation setting. Also, when constructing the prompt, we do not use the provided explanations for fair comparison. For ViLP, we consistently select `image3` and `answer3` to curate our (counterfactual) VQA pair. For VLind-Bench, we convert each annotated counterfactual statement into a binary 'Yes/No' question. To perform this transformation, we use the advanced language model `GPT-4o` with the following prompt:

> Generate a question based on the counterfactual information in the given statement. The question should be answered by yes.
> Here are some examples:
> Statement: The Statue of Liberty is holding a sword instead of a torch. Question: Is the Statue of Liberty holding a sword?
> Statement: The Sydney Opera House is illustrated as an underwater aquarium, with fish swimming around its structures. Question: Is the Sydney Opera House underwater?
> Statement: The Leaning Tower of Pisa is perfectly vertical in the image, without any tilt. Question: Is the Leaning Tower of Pisa perfectly vertical?
> Now generate a question for the following statement: {statement}

The models are instructed to directly generate the answer under a zero-shot setting, without involving any reasoning steps. Additional statistics of each dataset and corresponding splits are provided in Table 7.

To further validate the reliability of the agreement-based separation introduced in §3.2, and to demonstrate that our analysis can generalize to scenarios with known visual dependencies, we construct a controlled baseline dataset. Specifically, we take questions from CommonsenseQA (Talmor et al., 2018), which are inherently language-only, and pair each question with a randomly selected, irrelevant image from COCO 2017-val (Lin et al., 2014), forming a synthetic VQA setting that does not require visual input. We treat this as our vision-independent group $\mathcal{D}_T$. For the vision-

https://huggingface.co/llava-hf/llava-v1.6-vicuna-7b-hf
https://huggingface.co/llava-hf/llava-onevision-qwen2-7b-ov-hf
https://huggingface.co/OpenGVLab/InternVL3-8B-hf
https://huggingface.co/nvidia/Eagle2.5-8B
https://huggingface.co/google/gemma-3-4b-it
https://huggingface.co/google/gemma-3-12b-it
https://huggingface.co/google/gemma-3-27b-it
https://huggingface.co/meta-llama/Llama-3.2-11B-Vision-Instruct
https://huggingface.co/HuggingFaceTB/SmolVLM-Instruct
https://huggingface.co/Salesforce/instructblip-vicuna-7b

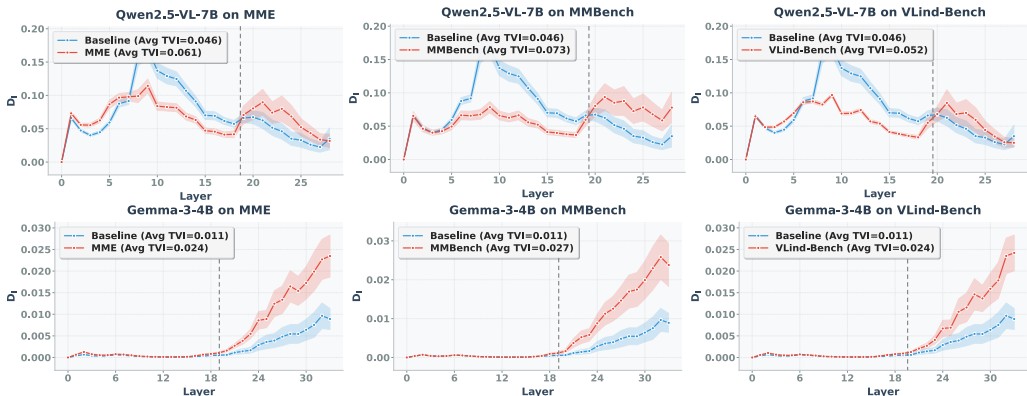

Figure 10: **Experimental results under controlled settings.** We use a synthetic baseline constructed from CommonsenseQA questions paired with irrelevant images as $\mathcal{D}_T$ (vision-independent), while standard VQA benchmarks (MME, MMBench, and VLind-Bench) are used as $\mathcal{D}_{VT}$ (vision-dependent).

dependent group $\mathcal{D}_{VT}$, we use samples from standard VQA benchmarks such as MME, MMBench, and VLind-Bench, which typically require more grounding in visual content. As shown in Figure 10, the average TVI is significantly lower for the baseline $\mathcal{D}_T$ compared to the general VQA datasets, confirming that the model does not extract meaningful information from irrelevant visual input. In contrast, even in the presence of strong language priors, the model still benefits from image content in $\mathcal{D}_{VT}$. It is also worth noting that there is a reverse trend before VIP for `Qwen2.5-VL-7B`, indicating representation distance before VIP is not associated with the actual meaningful visual integration during decoding. These findings are consistent with our previous results and further support the effectiveness of our proposed separation framework. Nonetheless, to ensure better control over data attributes such as format and context length, and to reveal clearer trends, we continue to use the agreement-based separation strategy in the main text.

Table 7: **Dataset statistics.** $M$ stands for multiple-choice and $B$ stands for binary-choice (Yes/No). `Qwen2.5-VL-7B` is used as an example here.

| Statistics | MME | MMBench | MMStar | MMMU | VLind-Bench | ViLP |
|---|---|---|---|---|---|---|
| Question Type | B | M | M | M | B | M |
| $|\mathcal{D}|$ | 2374 | 4377 | 1500 | 805 | 418 | 300 |
| $|\mathcal{D}_{VT}|$ | 546 | 2782 | 1057 | 446 | 144 | 177 |
| $|\mathcal{D}_T|$ | 1828 | 1595 | 443 | 359 | 274 | 123 |
| $|\mathcal{D}_{VT}|/|\mathcal{D}_T|$ | 0.30 | 1.74 | 2.39 | 1.24 | 0.53 | 1.44 |

**Metrics.** All TVI values reported in our experiments are computed based on empirically determined Visual Integration Points (VIPs) specific to each model. The following VIPs are used: `Qwen2.5-VL-7B` (18), `InternVL3-8B` (16), `Gemma-3-4B` (20), `Gemma-3-12B` (26), `Gemma-3-27B` (35), `LLaVA-v1.5-7B` (9), `Eagle2.5-8B` (15), `Llama-3.2-11B-Vision` (12), `LLaVA-NeXT-Vicuna-7B` (12), `LLaVA-OV-Qwen2-7B` (15) and `SmolVLM` (15)[6]. We also provide an automatic method for estimating VIP in Appendix B.2 for potential practical usage. Representation distances are computed using the hidden states corresponding to the last generated token. The metrics introduced in §4 are computed as follows:

$$\text{Visual Attention} = \frac{1}{LH}\sum_{l=1}^{L}\sum_{h=1}^{H}\alpha^{(l,h)}, \quad \text{Output Divergence} = d(Z_{\text{vis}}^{L}, Z_{\text{blind}}^{L}) \qquad (10)$$

---

[6]It should be noted that these manually selected VIPs are not necessarily optimal; however, they already achieve strong and robust effectiveness and are sufficient for analytical purpose. The truly optimal VIP is likely to lie in their vicinity.

Table 8: **Summary of evaluated models and their architectural specifications.**

| Model | Fusion | # Layers | Hidden Size |
|---|---|---|---|
| Qwen2.5-VL-7B/32B/72B | MLP | 28 / 64 / 80 | 3584 / 5120 / 8192 |
| Gemma-3-4B/12B/27B | Q-Former | 34 / 48 / 62 | 2560 / 3840 / 5376 |
| InternVL3-8B | MLP | 28 | 3584 |
| LLaVA-v1.5-7B | MLP | 32 | 4096 |
| Eagle2.5-8B | MLP | 28 | 3584 |
| Llama-3.2-11B-Vision | X-Attention | 40 | 4096 |
| LLaVA-NeXT-Vicuna-7B | MLP | 32 | 4096 |
| LLaVA-OV-Qwen2-7B | MLP | 28 | 3584 |
| SmolVLM | MLP | 24 | 2048 |
| InstructBLIP-Vicuna-7B | Q-Former | 32 | 4096 |

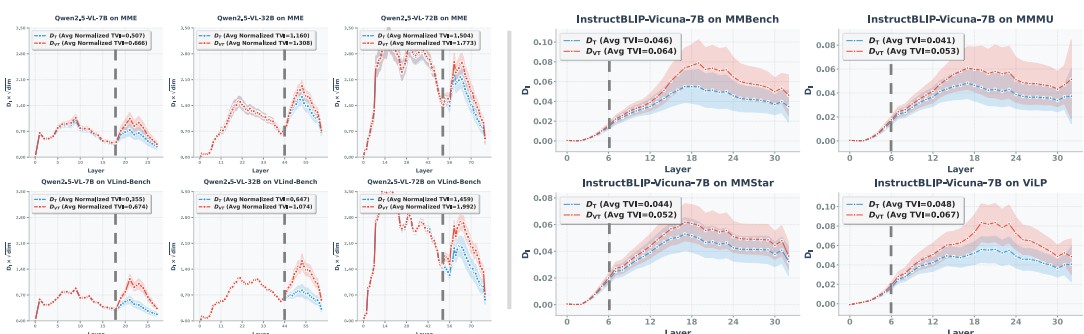

Figure 11: **Extended experiments across models of varying scales and fusion architectures.**

where $\alpha^{(l,h)}$ denotes the total attention from the final generated token to all preceding visual tokens in head $h$ at layer $l$, and $Z^L$ represents the hidden state at the final layer.

## D  FULL RESULTS OF THE MAIN EXPERIMENT

We provide the complete experimental results on 9 LVLMs and 6 datasets in Figure 6, 7, 8. Across all models and datasets, a consistent existence of the VIP can be observed. However, for some models, such as SmolVLM, the divergence between $\mathcal{D}_{VT}$ and $\mathcal{D}_T$ is less pronounced, likely due to the model's limited capacity and thus less promising visual integration.

**Additional architectures & scales.** We further validate our findings on models with diverse architectural designs, including Llama-3.2-11B-Vision with cross-attention–based multimodal fusion and InstructionBLIP-Vicuna-7B with a Q-Former–style fusion mechanism, as well as on larger-scale models such as Qwen2.5-VL-32B and Qwen2.5-VL-72B. As shown in Figure 11, the results consistently corroborate our conclusions across both architectural variants and model scales. Comprehensive statistics for all evaluated models are provided in Table 8.

## E  FURTHER ANALYSIS

**Analysis on the limitations of attention-based and output-based LP analysis.** First, for attention-based methods, we argue that a higher visual attention weight does not necessarily imply better visual grounding. As shown in Figure 12, under weak language priors, the model is able to correctly attend to the key areas that are semantically related to the given instruction. However, under strong language priors, we observe a pathological attention pattern in which the model's attention becomes abnormally concentrated in a limited region of the image. We refer to this phenomenon as an attention sink. In such cases, although the aggregated visual attention weight appears high, it does not reflect meaningful visual processing. Instead, the model is effectively bypassing genuine visual understanding by fixating on irrelevant or static regions, thereby undermining the utility of attention-based metrics as reliable indicators of visual integration.

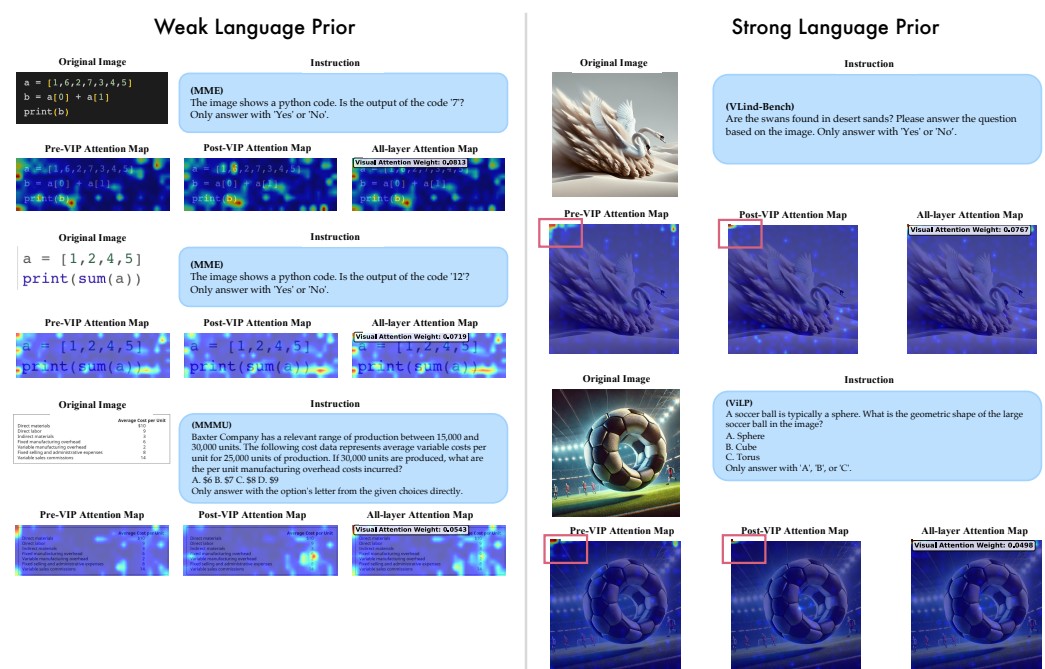

Figure 12: **Visualization of visual attention maps under weak and strong language priors.**

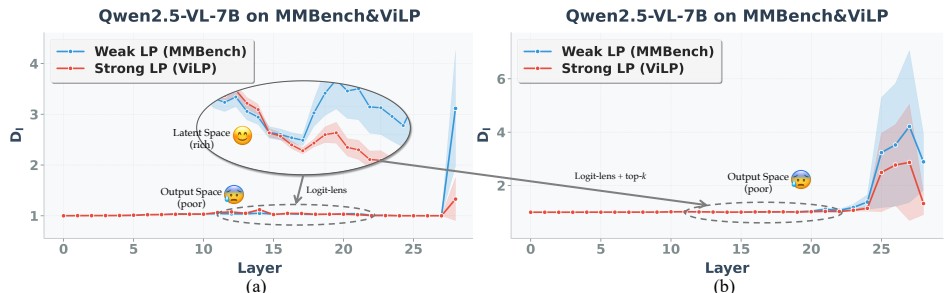

Figure 13: **Layer-wise representation distances in latent space vs. output space.** We apply the logit-lens to project hidden states at each layer into the output space. In (a), distances are computed over the entire output vector, while in (b), they are restricted to the top-$k$ token positions corresponding to candidate answer options.

To better understand why output-based representations are less effective in capturing language prior behavior, we visualize how representation distances vary across the latent and output spaces. As shown in Figure 13, the projection from the latent space to the output token space tends to obscure semantic distinctions that are otherwise indicative of the model's underlying behavior—such as whether it is performing effective visual grounding or defaulting to language priors. This observation aligns with our earlier argument: surface-level outputs alone may not faithfully reflect the internal decision-making process of LVLMs. Instead, meaningful behavioral signals often reside in the deeper latent representations, emphasizing the importance of analyzing internal dynamics rather than relying solely on output-level comparisons.

**Ablation on aggregation strategies for TVI calculation.** In our main experiments, we adopt a simple aggregation strategy for computing TVI by averaging the representation distances across all post-VIP layers. To assess whether more sophisticated aggregation may improve the metric, we

additionally experiment with a standard-deviation-based weighted TVI, defined as:

$$\text{TVI}(l^*; x, F_\theta) = \frac{1}{L - l^* + 1} \sum_{l=l^*}^{L} \left[ \sigma_l \cdot d(z_{\text{vis}}^l, z_{\text{blind}}^l) \right], \quad (11)$$

where $\sigma_l$ denotes the standard deviation of representation distances at layer $l$. This weighting scheme emphasizes layers whose distance distributions exhibit higher variability, i.e., layers that are more sample-specific and potentially more informative, while down-weighting contributions that may arise from less discriminative layers. As shown in Table 9, this reweighting provides a slight improvement in effectiveness over the simple aver-

Table 9: **Comparison of aggregation methods for computing TVI.**

| Model | Aggregation | VLind | ViLP |
|---|---|---|---|
| Qwen2.5-VL-7B | Simple Average | 0.7155 | 0.6335 |
| Qwen2.5-VL-7B | Std Reweighting | 0.7164 | 0.6348 |
| InternVL3-8B | Simple Average | 0.6727 | 0.5709 |
| InternVL3-8B | Std Reweighting | 0.6739 | 0.5723 |

aging strategy across both VLind and ViLP. However, the gains are modest and come at the cost of additional computation. These findings suggest that our original simple averaging approach already serves as a strong and robust summary statistic, while the weighted variant may offer incremental refinement in specialized scenarios.

**Analysis on instruction-level perturbation.** To strengthen our study of the contrasting chain-of-embedding, we compare embedding chains produced by image-text inputs versus image-only inputs (Figure 14 (a)). As expected, the two curves exhibit only negligible differences, likely due to the limited semantic content of image-only inputs. This phenomenon can be attributed to two main factors: (1) The hidden states from image-only inputs do not reveal anything about the model's answer prediction, considering that the instructions are not even provided. The resulting representations thus are not directly comparable in the same behavioral space. (2) Most current LVLMs are not trained to handle

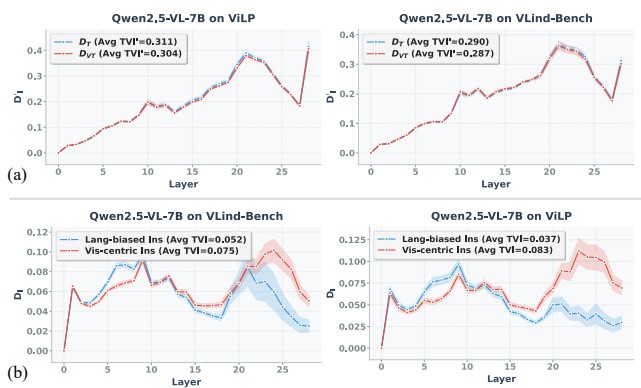

Figure 14: **Impacts of instruction-level perturbation.** (a) Representation distances obtained by contrasting chain-of-embedding sequences produced *with and without textual instructions.* (b) TVI scores under instructions with *different styles*

image-only inputs for question answering, which makes the model's behavior on image-only inputs unpredictable. Nonetheless, even under these constraints, we still observe that the revised TVI, which now specifically captures the contribution of textual content to the decoding process, is marginally higher for vision-independent samples than for vision-dependent ones. This further corroborates the robustness of our analysis framework and the consistency of our conclusions.

To further isolate the influence of the textual component in multimodal inputs, we investigate how different instruction styles affect model behavior. Specifically, we contrast misleading instructions (those originating from VLind-Bench or ViLP that elicit strong LP) with generic, vision-centric prompts such as "Describe the image in detail." As results in Figure 14 (b) show, TVI under these vision-centric instructions is significantly higher than under misleading ones. This confirms that TVI is sensitive to the model's bias toward language priors under different instruction regimes.

**Analysis on VIP & TVI across different training stages.** To better understand how VIP and TVI evolve during model training, we analyze checkpoints from multiple stages of LLaVA-v1.5 visual instruction tuning. As shown in Figure 15 and Figure 16, our evaluation reveals two key

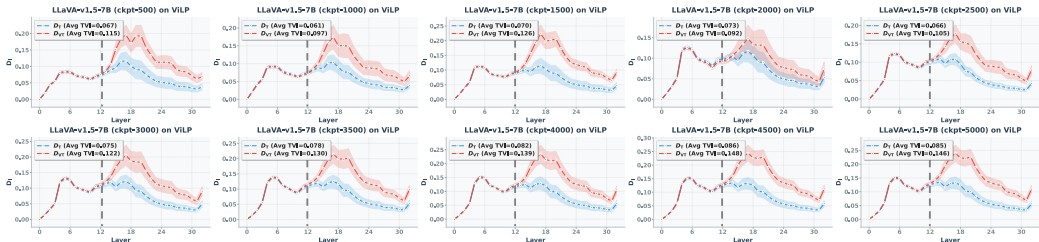

Figure 15: **Evaluation results across different stages of LLaVA-v1.5's visual instruction tuning.**

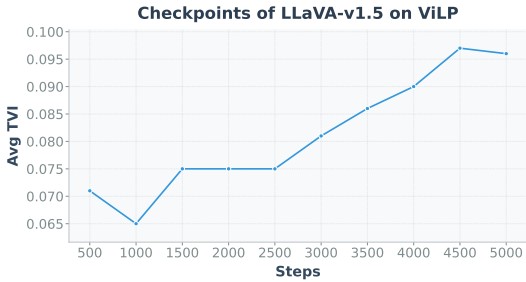

Figure 16: **Average TVI on ViLP across different stages of visual instruction tuning.**

observations. First, the VIP position remains remarkably stable across training stages (around the 12th layer). This suggests that the mechanism governing where visual information is integrated is largely established during pretraining and persists throughout subsequent finetuning. In other words, VIP appears to reflect an intrinsic architectural property rather than a behavior shaped by instruction tuning. Second, TVI exhibits a clear upward trend as training progresses, indicating that the degree to which the model effectively incorporates visual information improves gradually during instruction tuning. These findings align with our previous conclusion that VIP is a model-specific property as well as our expectation that visual instruction tuning incrementally enhances the model's multimodal fusion capability.

**Analysis on when TVI fails.** Although TVI shows strong correlation with answer correctness as shown in Table 3, there are still some non-neglectable portion on failure cases that TVI can not properly predict the model's answer quality. Conceptually, TVI is designed to capture the degree of effective visual integration during decoding. While higher TVI generally implies that the generated answer is more grounded in visual input and thus more likely to be correct, there are natural regimes where this relationship weakens, for example: (1) Some questions are not strongly visually demanding; the model can answer them correctly with relatively little visual integration. In such cases, the answer may be correct even when TVI is small, leading to "false negatives" from TVI's perspective. (2) For very challenging visual questions (e.g., fine-grained recognition, subtle spatial reasoning, or interpreting small/occluded objects) the model may still fail even after substantial effort to integrate visual information (high TVI). This yields "false positives" in terms of TVI–correctness alignment. These factors introduce inherent noise into any evaluation based on TVI vs. correctness.

To further characterize these patterns, we visualize the TVI distribution for ViLP samples that Qwen2.5-VL-7B answers correctly and perform a focused analysis on those with unexpectedly low TVI. As shown in Figure 17, while the majority of correctly answered samples exhibit relatively high TVI, a small subset attains low TVI despite being answered correctly. Upon inspection, many of these cases can indeed be solved without relying on visual input, either due to knowledge leakage or lucky guessing. This suggests that such questions are intrinsically less visually demanding, and the low TVI observed in these cases does not constitute a failure of the metric but instead reflects the fundamental properties of the task itself.

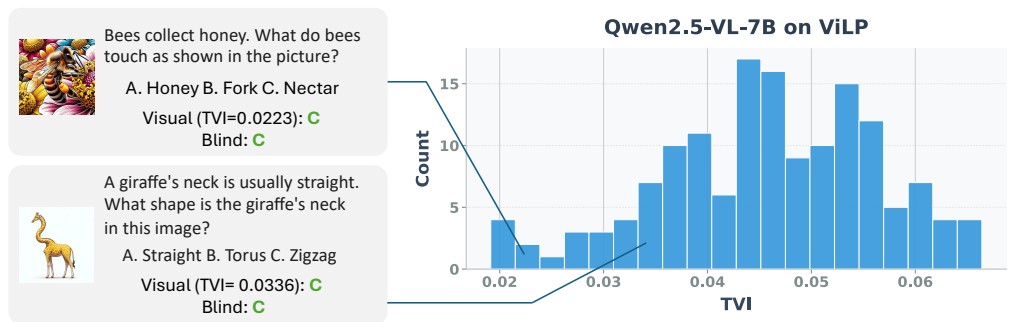

Figure 17: **Visualization of TVI distribution and case studies on TVI failure cases.**

# F  DETAILS ON THEORETICAL ANALYSIS AND PROOFS

## F.1  JUSTIFICATION AND INTERPRETATION ON REPRESENTATION DIVERGENCE

We first show that our empirical estimate for the difference in the expected representation distances (Eq. 3) can be viewed as a two-sample test statistic with asymptotic normality in Lemma F.1.

**Lemma F.1.** *Let $X = (X_v, X_t) \in \mathcal{X}$ be a random variable sampled from $\mathcal{P}_{VT}$ or $\mathcal{P}_T$, and denote $\mathcal{D}_{VT} \sim \mathcal{P}_{VT}$ and $\mathcal{D}_T \sim \mathcal{P}_T$ as empirical distributions with $N$ and $M$ i.i.d. samples, respectively. Given a stack of LVLM layers $f_l : \mathcal{X} \to \mathcal{Z}$ from $F_\theta$ and a distance metric $d : \mathcal{Z} \times \mathcal{Z} \to \mathbb{R}$ with a finite second moment, the difference in the expected representation distance estimates $\mathbf{D}_l(\cdot, \cdot)$ between $\mathcal{D}_{VT}$ and $\mathcal{D}_T$ is a two-sample test statistic with asymptotic normality, that is,*

$$\mathbf{D}_l(\mathcal{D}_{TV}, F_\theta) - \mathbf{D}_l(\mathcal{D}_T, F_\theta) \overset{\text{approx}}{\sim} \mathcal{N}(\mu_T - \mu_{VT}, \frac{\sigma_T^2}{M} + \frac{\sigma_{VT}^2}{N}) \quad as \ \ N, M \to \infty, \qquad (12)$$

*where $\mu_{VT} = \mathbf{D}_l(\mathcal{P}_{VT}, F_\theta)$, $\mu_T = \mathbf{D}_l(\mathcal{P}_T, F_\theta)$ and $\sigma_{VT}^2 = Var_{\mathcal{P}_{VT}}[d(f_l(X_v, X_t), f_l(X_t))] < \infty$, $\sigma_T^2 = Var_{\mathcal{P}_T}[d(f_l(X_v, X_t), f_l(X_t))] < \infty$.*

*Proof.* Let $\mathcal{I}_{VT}$ and $\mathcal{J}_T$ be the index sets of $N$ and $M$ i.i.d. samples from $\mathcal{P}_{VT}$ and $\mathcal{P}_T$, respectively. If samples from $\mathcal{P}_{VT}$ and $\mathcal{P}_T$ are independent, with finite second moments, we have,

$$\mathbf{D}_l(\mathcal{D}_{VT}, F_\theta) - \mathbf{D}_l(\mathcal{D}_T, F_\theta)$$
$$= \frac{\sum_{i \in \mathcal{I}_{VT}}[d(f_l(x_v^i, x_t^i), f_l(x_t^i))]}{N} - \frac{\sum_{j \in \mathcal{J}_T}[d(f_l(x_v^j, x_t^j), f_l(x_t^j))]}{M} \qquad (13)$$
$$\sim \mathcal{N}(\mu_T - \mu_{VT}, \frac{\sigma_T^2}{M} + \frac{\sigma_{VT}^2}{N}) \qquad \text{(by Central Limit Theorem as } N, M \to \infty)$$

Note that, even though the samples from $\mathcal{P}_{VT}$ and $\mathcal{P}_T$ are not independent, the asymptotic normality still holds by considering covariance terms between the two distributions. □

Now, in Lemma F.2, we provide a new interpretation of our representation distance measure between CoE by casting the distance measurement as a density estimation problem.

**Lemma F.2.** *Let $X = (X_v, X_t) \in \mathcal{X}$ be a random variable sampled from $\mathcal{P}_{VT}$ or $\mathcal{P}_T$, and $f_l : \mathcal{X} \to \mathcal{Z}$ be a stack of LVLM layers. For $\mathcal{P}_T$, define a density estimator $\hat{p}_T(Z^l) := \mathcal{N}(Z^l; f_l(X_t), I)$. Given a squared $l_2$ distance $d(Z_1, Z_2) := \frac{1}{2}\|Z_1 - Z_2\|_2^2$, the representation distance $d(f_l(X_v, X_t), f_l(X_t))$ is the negative log-likelihood estimate of $Z^l$ from $\mathcal{P}_T$, denoted as $\hat{p}_T(Z^l)$, up to an additive constant. That is,*

$$d(f_l(X_v, X_t), f_l(X_t)) \triangleq -\log \hat{p}_T(Z^l) + \log C, \qquad (14)$$

*where $C = (2\pi)^{-\frac{d_z}{2}}$ is a normalizing constant for the $d_z$-dimensional unit-variance Gaussian.*

*Proof.* It is easy to check,

$$\hat{p}_{\mathrm{T}}(Z^l) = \mathcal{N}(Z^l; Z^l_{\mathrm{blind}}, I) \tag{15}$$

$$\hat{p}_{\mathrm{T}}(Z^l) = C \cdot \exp(-\frac{||Z^l - Z^l_{\mathrm{blind}})||_2^2}{2}) \tag{16}$$

$$\log \hat{p}_{\mathrm{T}}(Z^l) = \log C - \frac{||Z^l - Z^l_{\mathrm{blind}}||_2^2}{2} \tag{17}$$

$$\log \hat{p}_{\mathrm{T}}(Z^l) = \log C - d(Z^l, Z^l_{\mathrm{blind}}) \tag{18}$$

where $Z_{\mathrm{blind}} = f_l(X_t)$. $\square$

In other words, the distance between the original representation $Z^l = f_l(X_v, X_t)$ and the blind one $Z^l_{\mathrm{blind}} = f_l(X_t)$ can be expressed as a probability density estimate of $Z^l$ given $\mathcal{N}(\cdot; Z^l_{\mathrm{blind}}, I)$, which has a mean $Z^l_{\mathrm{blind}}$ with the isotropic variance, as our estimator. On top of this new framing, *i.e.*, distance measurement as a density estimation problem, we further provide an information-theoretic interpretation of the difference in expected representation distance in Theorem F.3.

**Theorem F.3** (Restatement of Theorem 5.1). *Let $X = (X_v, X_t) \in \mathcal{X}$ be a random variable from $\mathcal{P}_{VT}$ or $\mathcal{P}_T$, and $f_l : \mathcal{X} \to \mathcal{Z}$ be a layer stack from an LVLM $F_\theta$. For $\mathcal{P}_T$, define a density estimator $\hat{p}_T(Z^l) := \mathcal{N}(Z^l; f_l(X_t), I)$, and denote $p_{VT}(Z^l)$ (resp. $p_T(Z^l)$) as the population distribution on $Z^l = f_l(X)$ derived from $\mathcal{P}_{VT}$ (resp. $\mathcal{P}_T$). Then, given $d(Z_1, Z_2) := \frac{1}{2}||Z_1 - Z_2||_2^2$, the difference in the expected representation distances between $\mathcal{P}_{VT}$ and $\mathcal{P}_T$, i.e., $\mathbf{D}_l(\mathcal{P}_{VT}, F_\theta) - \mathbf{D}_l(\mathcal{P}_T, F_\theta)$, can be expressed as follows,*

$$KL\big(p_{VT}(Z^l)||\hat{p}_T(Z^l)\big) - KL\big(p_T(Z^l)||\hat{p}_T(Z^l)\big) + \bar{\mathbf{H}}, \tag{19}$$

*where $\bar{\mathbf{H}}$ is a constant $H\big(p_{VT}(Z^l)\big) - H\big(p_T(Z^l)\big)$, and $KL(\cdot||\cdot)$ denotes the KL divergence.*

*Proof.*

$$\mathbf{D}_l(\mathcal{P}_{VT}, F_\theta) - \mathbf{D}_l(\mathcal{P}_T, F_\theta)$$

$$= \mathbb{E}_{\mathcal{P}_{VT}}[d(f_l(X_v, X_t), f_l(X_t))] - \mathbb{E}_{\mathcal{P}_T}[d(f_l(X_v, X_t), f_l(X_t))] \tag{20}$$

$$= \mathbb{E}_{p_{VT}(Z^l)}[-\log \hat{p}_T(Z^l) + \log C] - \mathbb{E}_{p_T(Z^l)}[-\log \hat{p}_T(Z^l) + \log C] \tag{21}$$

$$= \mathbb{E}_{p_{VT}(Z^l)}[-\log \hat{p}_T(Z^l)] - \mathbb{E}_{p_T(Z^l)}[-\log \hat{p}_T(Z^l)] \tag{22}$$

$$= H\big(p_{VT}(Z^l), \hat{p}_T(Z^l)\big) - H\big(p_T(Z^l), \hat{p}_T(Z^l)\big) \tag{23}$$

$$= \big[\mathbf{KL}\big(p_{VT}(Z^l)||\hat{p}_T(Z^l)\big) + H\big(p_{VT}(Z^l)\big)\big] - \big[\mathbf{KL}\big(p_T(Z^l)||\hat{p}_T(Z^l)\big)\big] + H\big(p_T(Z^l)\big)\big] \tag{24}$$

where $H(\cdot)$ and $H(\cdot, \cdot)$ denote entropy and cross-entropy, respectively. Here, Eq. 21 holds by Lemma F.2, and the remaining equality is trivial by the definitions of $d()$ and information theoretic measures. $\square$

This simple theorem gives us a new interpretation on the measure of representation divergence, $\mathbf{D}_l(\mathcal{P}_{VT}, F_\theta) - \mathbf{D}_l(\mathcal{P}_T, F_\theta)$: the amount of expected excess surprisal when we assume that the sample representation follows blind representation-centered normal distribution compared to the true population distribution $\mathcal{P}_{VT}$, compensated by estimation quality $\mathrm{KL}\big(p_T(Z^l)||\hat{p}_T(Z^l)\big)$.

## F.2 Notations and Problem Setup for Theorem F.6

We recast the problem of measuring the representation distance $d(Z_{\mathrm{vis}}, Z_{\mathrm{blind}})$ as a binary classification task, where we want to classify the sample $(Z_{\mathrm{vis}}, Z_{\mathrm{blind}})$ into 1 if it originates from the distribution $\mathcal{P}_{VT}$ while 0 for the samples from $\mathcal{P}_T$.

To be specific, let $\mathcal{X}$, $\mathcal{Z}$, and $\mathcal{Y}$ denote input, LVLM representation, and output, respectively. We have a multimodal input query $X = (X_v, X_t) \in \mathcal{X}$, a stack of LVLM layers $f_l : \mathcal{X} \to \mathcal{Z}$, and a distance metric $d : \mathcal{Z} \times \mathcal{Z} \to [0, 1]$. With that, we define a hypothesis $h = d(f_l(X_v, X_t), f_l(X_t)) : \mathcal{X} \to [0, 1]$ as a real value function to measure the relative likelihood that the input $X$ is sampled from $\mathcal{P}_{VT}$ rather than $\mathcal{P}_T$, and we also define the labeling function $h^\star : \mathcal{X} \to \{0, 1\}$ that maps the input into its ground-truth membership, *i.e.*, 1 if it's from $\mathcal{P}_{VT}$ and 0 if it's from $\mathcal{P}_T$. Then,

we formulate an expected error (a.k.a. *risk*) of a hypothesis $h$ w.r.t. the labeling function $h^\star$ on a distribution $\mathcal{P}$ as follows: $\varepsilon_{\mathcal{P}}(h, h^\star) := \mathbb{E}_{X \sim \mathcal{P}}[|h(X) - h^\star(X)|]$.

Besides, in Def. F.4, we introduce a measure of discrepancy between two distributions, $\mathcal{H}$-divergence, which has been widely adopted in domain adaptation literature (Ben-David et al., 2006; 2010; Ganin et al., 2016; Zhao et al., 2018), and also VLM fine-tuning regimes (Oh et al., 2024).

**Definition F.4** ($\mathcal{H}$-divergence, (Ben-David et al., 2006; 2010)). *Let $\mathcal{P}$ and $\mathcal{P}'$ be probability distributions on the input domain $\mathcal{X}$, and $\mathcal{H}$ be a hypothesis class for $\mathcal{X}$. Denote $\mathcal{A}_{\mathcal{H}} := \{h^{-1}(1) | h \in \mathcal{H}\}$ as a collection of subsets of $\mathcal{X}$ that are the support of some hypotheses in $\mathcal{H}$. Then, the distance between $\mathcal{P}$ and $\mathcal{P}'$ based on $\mathcal{H}$ is defined as follows:*

$$d_{\mathcal{H}}(\mathcal{P}, \mathcal{P}') = 2 \sup_{A \in \mathcal{A}_{\mathcal{H}}} |\mathbb{P}_{\mathcal{P}}[A] - \mathbb{P}_{\mathcal{P}'}[A]|. \tag{25}$$

Now, we are ready to present the proof for Proposition 5.2 in the next subsection.

### F.3   PROOF FOR THEOREM F.6

**Theorem F.5** (Restatement of Theorem 5.2). *Let $X = (X_v, X_t) \in \mathcal{X}$ be a random variable of a multimodal input query. Given a stack of LVLM layers $f_l : \mathcal{X} \to \mathcal{Z}$ and a distance metric $d : \mathcal{Z} \times \mathcal{Z} \to [0, 1]$, define a hypothesis $h = d(f_l(X_v, X_t), f_l(X_t)) : \mathcal{X} \to [0, 1]$ and a set of these hypotheses $\mathcal{H}$ that has a pseudo-dimension $c$. Then, for $\mathbf{D}_l(\mathcal{P}_\star, F_\theta) := \mathbb{E}_{X \sim \mathcal{P}_\star}[h(X)]$ with any $\mathcal{P}_{VT}, \mathcal{P}_T$, and $\mathcal{P}_M := \frac{\mathcal{P}_{VT} + \mathcal{P}_T}{2}$, and the empirical distributions $\mathcal{D}_{VT} \sim \mathcal{P}_{VT}$ and $\mathcal{D}_T \sim \mathcal{P}_T$ of $N$ samples for each, we have the following bounds w.p. at least $1 - \delta$ for $0 < \delta < 1$,*

$$i) \ \ 1 - \mathbf{D}_l(\mathcal{D}_T, F_\theta) - \frac{1}{2} d_{\bar{\mathcal{H}}}(\mathcal{D}_{VT}, \mathcal{D}_T) - \tilde{\mathcal{O}}_\delta \leq \mathbf{D}_l(\mathcal{P}_{VT}, F_\theta), \tag{26}$$

$$ii) \ \ \frac{1}{2} - \frac{1}{4} d_{\bar{\mathcal{H}}}(\mathcal{D}_{VT}, \mathcal{D}_T) - \tilde{\mathcal{O}}_\delta \leq \mathbf{D}_l(\mathcal{P}_M, F_\theta) \leq \frac{1}{2} + \frac{1}{4} d_{\bar{\mathcal{H}}}(\mathcal{D}_{VT}, \mathcal{D}_T) + \tilde{\mathcal{O}}_\delta \tag{27}$$

*where $\bar{\mathcal{H}} := \{\mathbb{I}_{|h(X) - h'(X)| > t} : h, h' \in \mathcal{H}, 0 \leq t \leq 1\}$ and $\tilde{\mathcal{O}}_\delta := \mathcal{O}(\sqrt{\frac{1}{N}(\log \frac{1}{\delta} + c \log \frac{N}{c})})$.*

*Proof.* Note the lemma below that provides a connection between the difference in the expected errors across two distributions and their distributional discrepancy.

**Lemma F.6** (Zhao et al. (2018)). *For $h, h' \in \mathcal{H} := \{h : \mathcal{X} \to [0, 1]\}$ assume that $\mathcal{H}$ has a finite pseudo dimension $d$. For any distribution $\mathcal{P}$ and $\mathcal{P}'$ over $\mathcal{X}$,*

$$|\varepsilon_{\mathcal{P}}(h, h') - \varepsilon_{\mathcal{P}'}(h, h')| \leq \frac{1}{2} d_{\bar{\mathcal{H}}}(\mathcal{P}, \mathcal{P}'), \tag{28}$$

*where $\bar{\mathcal{H}} := \{\mathbb{I}_{|h(x) - h'(x)| > t} : h, h' \in \mathcal{H}, 0 \leq t \leq 1\}$.*

See Lemma 1 of Zhao et al. (2018) for the proof. We start our derivation of Proposition 5.2 from the ineq. F.6 as below,

$$\frac{1}{2} d_{\bar{\mathcal{H}}}(\mathcal{P}_{VT}, \mathcal{P}_T) \geq |\varepsilon_{\mathcal{P}_{VT}}(h, h^\star) - \varepsilon_{\mathcal{P}_T}(h, h^\star)| \tag{29}$$

$$= \left| |\mathbf{D}_l(\mathcal{P}_{VT}, F_\theta) - 1| - |\mathbf{D}_l(\mathcal{P}_T, F_\theta) - 0| \right| \tag{30}$$

$$= |1 - \mathbf{D}_l(\mathcal{P}_{VT}, F_\theta) - \mathbf{D}_l(\mathcal{P}_T, F_\theta)| \tag{31}$$

$$\geq |1 - \mathbf{D}_l(\mathcal{P}_{VT}, F_\theta)| - |\mathbf{D}_l(\mathcal{P}_T, F_\theta)| \tag{32}$$

$$= 1 - \mathbf{D}_l(\mathcal{P}_{VT}, F_\theta) - \mathbf{D}_l(\mathcal{P}_T, F_\theta), \tag{33}$$

where the first equality holds by definition, the first inequality holds by the reverse triangular inequality, and the second and fourth equality hold given $0 \leq \mathbf{D}_l(\mathcal{P}_\star, F_\theta) \leq 1$.

In the meantime, for the empirical distributions $\mathcal{D}_{VT} \sim \mathcal{P}_{VT}$ and $\mathcal{D}_T \sim \mathcal{P}_T$ of $N$ samples for each, given $0 < \delta < 1$, we have the following approximation error bounds with probability at least $1 - \delta$

for any $h \in \mathcal{H}$ (See Lemma 5 and Lemma 6 of Zhao et al. (2018)),

$$\varepsilon_{\mathcal{P}_\star}(h, h^\star) \leq \varepsilon_{\mathcal{D}_\star}(h, h^\star) + \mathcal{O}(\sqrt{\frac{1}{N}(\log\frac{1}{\delta} + c\log\frac{N}{c})}), \tag{34}$$

$$d_{\bar{\mathcal{H}}}(\mathcal{P}_{\mathrm{VT}}, \mathcal{P}_{\mathrm{T}}) \leq d_{\bar{\mathcal{H}}}(\mathcal{D}_{\mathrm{VT}}, \mathcal{D}_{\mathrm{T}}) + \mathcal{O}(\sqrt{\frac{1}{N}(\log\frac{1}{\delta} + c\log\frac{N}{c})}), \tag{35}$$

where $Pdim(\mathcal{H}) = c$.

Then, by plugging the above inequality (Ineq. 35) into the Ineq. 31, we have,

$$1 - \frac{1}{2}d_{\bar{\mathcal{H}}}(\mathcal{D}_{\mathrm{VT}}, \mathcal{D}_{\mathrm{T}}) - \mathcal{O}(\sqrt{\frac{1}{N}(\log\frac{1}{\delta} + c\log\frac{N}{c})}) \leq \mathbf{D}_l(\mathcal{P}_{\mathrm{VT}}, F_\theta) + \mathbf{D}_l(\mathcal{P}_{\mathrm{T}}, F_\theta), \tag{36}$$

$$1 + \frac{1}{2}d_{\bar{\mathcal{H}}}(\mathcal{D}_{\mathrm{VT}}, \mathcal{D}_{\mathrm{T}}) + \mathcal{O}(\sqrt{\frac{1}{N}(\log\frac{1}{\delta} + c\log\frac{N}{c})}) \geq \mathbf{D}_l(\mathcal{P}_{\mathrm{VT}}, F_\theta) + \mathbf{D}_l(\mathcal{P}_{\mathrm{T}}, F_\theta), \tag{37}$$

where we derive the first statement of Proposition F.5 from the Ineq. 36, and the second statement of that by combining both Ineq. 36 and Ineq. 37, that complete the proof.

$\square$

## G  IMPACT STATEMENT

Language prior represents a pathological behavioral pattern in LVLMs, where the model overly relies on its linguistic knowledge and fails to properly ground its predictions in the visual input. This phenomenon underlies critical issues such as hallucination, modality misalignment, and failure cases in vision-centric reasoning. It also suggests that current LVLMs may not be operating in the modality-aware manner we expect—even when their outputs appear plausible (as the result of the vanilla next-token-prediction training paradigm). One of the main challenges in mitigating language prior lies in its vague and subjective nature: there exists no clear definition or quantitative measure of "language prior" in a dataset or task. Consequently, efforts to balance visual and textual information during training or fine-tuning often rely on heuristics or manual annotations.

Our work sheds light on this issue by proposing a formal framework to characterize and quantify the language prior through the model's own behavior. This makes the problem not only more visible but also more measurable. If the degree of language prior can be reliably estimated from within the model, we can begin to incorporate this signal directly into training objectives or inference strategies in a principled way. In this way, our framework provides a principled foundation for deeper understanding and offers practical tools for improving real-world multimodal systems.

In addition to the ultimate goal, i.e., understanding and quantifying LP of LVLM, our novel method, *contrastive chain-of-embeddings*, on the path to pursue that goal can also create a rich inspiration for a line of works on layer-wise representation analysis (Skean et al., 2025), layer-specific adaptive training approach for LVLMs (Bachu et al., 2025; Oh et al., 2025b), and inclusive AI applications with unbiased multimodal alignment (Kim et al., 2025) or representation-centric multi-linguality (Jung et al., 2024; Schut et al., 2025), which ultimately contribute to building a trustworthy multimodal AI system for everyone.

## H  DISCLOSURE OF LLM USAGE

Some portions of this paper were polished and refined with the assistance of LLM tools (*e.g.*, Chat-GPT) to improve clarity, fluency, and consistency in writing. We also harnessed a coding agent (*e.g.*, Cursor) to write some simple utility functions after double-checking. All technical content, experimental results, and analytical conclusions were independently developed by the authors without the use of LLMs.

## I  ETHICS STATEMENT

This work conducts empirical and analytical studies on the internal behavior of LVLMs, with the goal of understanding and quantifying their reliance on language priors and the extent of visual

information integration during inference. To pursue high standards of scientific excellence, we propose a formal framework with clear definitions of all used terms and conduct validation at scale, *e.g.*, 60 combinations of models and datasets, and we further provide theoretical analyses on our framework. Our study does not involve any human subjects, personally identifiable information, or sensitive data. All experiments are conducted using publicly available models and benchmark datasets that are widely adopted in the multimodal learning community. Our proposed metrics and analyses are intended for research and diagnostic purposes. By providing tools to diagnose when LVLMs rely on text versus vision, we aim to support more accountable model development and contribute positively to the responsible advancement of AI. We encourage future work to further validate these findings under more diverse real-world conditions.

## J    REPRODUCIBILITY STATEMENT

All of the models and datasets we used in this work are publicly available. To further ensure the reproducibility of our findings, we provide comprehensive descriptions of all experimental settings, including dataset preprocessing, model configurations, metric definitions, and evaluation protocols, in Section 4 and Appendix C. Our framework does not require model re-training or fine-tuning, and all evaluations are conducted in a zero-shot setting using publicly available model checkpoints, which minimizes computational and hardware requirements. We release the complete codebase for our analysis framework, including tools for data preparation, TVI computation, and visualization, at the following repository: ⍾.

