# OpenReview forum: "Understanding Language Prior of LVLMs by Contrasting Chain-of-Embedding"
_ICLR.cc/2026/Conference — ICLR 2026 Poster_

### Official Review · Reviewer_mW7M · 2025-10-31

**Soundness:** 2
**Presentation:** 2
**Contribution:** 3
**Rating:** 4
**Confidence:** 3

**Summary:**

This paper begins with reporting their observation of Visual Integration Point (VIP), the layer where visual input starts to affect hidden representations. Based on this observation, the paper proposes Total Visual Integration (TVI), a metric quantifying visual grounding (and, inversely, language-prior strength). TVI is distinct from prior approaches that rely on output comparisons using carefully curated datasets. Across nine LVLMs and six benchmark datasets, the authors find that VIP typically emerges at a similar depth of the decoder (agnostic to models) and that higher TVI correlates with better accuracy on vision-dependent samples. The authors also show that TVI outperforms previously proposed metrics like visual attention and output divergence in predicting model correctness on visually grounded tasks. Finally, they argue that TVI provides a more direct and interpretable measure of an LVLM’s visual grounding.

**Strengths:**

- The paper goes beyond conventional output level evaluations of language priors by examining layer-level hidden representations. It provides a new and interpretable perspective on how LVLMs integrate visual information and develop visual grounding.
- The newly proposed metric, TVI, offers a more robust and empirically and theoretically grounded way to quantify visual grounding (or, inversely, language prior strength) than prior metrics such as visual attention or output divergence. It directly measures the degree of vision’s influence on internal representations.
- The approach is easy to implement (requiring only paired forward passes with and without images) while yielding illuminating insights into when LVLMs begin to use visual input.
- The study evaluates nine LVLMs across six benchmarks, demonstrating that the observed VIP and TVI behaviors are overall consistent and generalizable across different architectures, dataset types, and model scales.

**Weaknesses:**

- The identification of VIP relies on dataset-level statistics and a tau threshold derived from pre-VIP variance, which introduces potential dependence on dataset composition (eg, the ratio of visually-dependent to -independent samples). The paper does not analyze how sensitive VIP detection is to this choice or whether tau threshold could be defined in a dataset-agnostic manner.

- Most of the 9 evaluated LVLMs use variations of late-fusion (adapter-based) architectures, with the exception of Gemma. This architectural bias may partly explain why the detected VIP consistently appears around the middle or later layers. The paper would be stronger if it included models with more diverse fusion architectures (eg, Flamingo-style, Q-Former, or Perceiver-based) to test whether or not the VIP phenomenon generalizes beyond late-fusion setups.

- It would also be valuable to test larger models. While API models cannot be easily analyzed, several powerful open-source LVLMs (e.g., 70B-scale models) could be evaluated to examine whether the observed trends in VIP and TVI scale to higher capacity models.

- In Table 2, TVI clearly outperforms other metrics, with correlation values ranging from 0.57 to 0.71. However, this still leaves a notable fraction of cases where TVI fails to capture visual grounding. An analysis of these failure patterns or outlier samples would have provided deeper insight into when TVI succeeds or breaks down, further strengthening the validity of the TVI metric.

- [Minor] While VIP and TVI are conceptually interesting, the paper does not clearly articulate how these metrics could guide future LVLM development. Including concrete examples of potential applications would make the work more actionable.

- [Minor] Although the paper emphasizes the number of evaluated models and benchmarks, these are not well-documented, even in the appendices. I recommend adding summary tables for both datasets and models. For models, include details such as fusion strategy and number of attention layers.

**Questions:**

See the above weaknesses.

Besides, while all evaluated models are relatively small, they still span a range of parameter scales (2B to 27B). Do the authors observe any trend in VIP location or TVI magnitude with respect to model size? It may be difficult to compare across different architectures, but evaluating models of varying sizes within the same family in future work could help clarify this relationship.

---

> ### Author Response · Authors · 2025-11-18
> **Response to Reviewer mW7M (1/3)**
>
> We sincerely thank the reviewer for acknowledging the strengths of our work. Besides, we really appreciate your constructive suggestions for technical clarification, experiments, and further analyses, which have greatly helped us to improve our work. We carefully reviewed all of them and respond as follows.
>
> > _W1. The identification of VIP relies on dataset-level statistics and a tau threshold derived from pre-VIP variance, which introduces potential dependence on dataset composition. The paper does not analyze how sensitive VIP detection is to this choice or whether tau threshold could be defined in a dataset-agnostic manner._
>
> We appreciate your concern. For clarification, we observe that **VIP (i.e., the layer where representation distances on $\mathcal{D}\_\text{VT}$ and $\mathcal{D}\_\text{T}$ begin to diverge) is indeed dataset-agnostic**, and is consistent across different datasets for each model. This is highlighted in `lines 224-227` and `Figure 3, 6, 7, 8` of our manuscript. We have also provide detailed statistics of each dataset used in `Table 7` (`Appendix C`), which further verify that **VIP is insensitive to dataset composition**.
>
> $\tau$ and VIPs used in our original manuscript are manually selected based on the observation on the empirical results of contrasting chain-of-embedding (ones like `Figure 3`). We provided an explanation of this process and the VIPs used for each model in `lines 1224-1230`, `Appendix C`. Below, we also show **how this process can be automated in a dataset-agnostic manner**.
>
> Specifically, **we implemented the automatic VIP detection algorithm** by taking inspiration from our theoretical analysis (Lemma F.1) showing that our representation divergence term $\mathbf{D}\_{l}(\mathcal{D}\_{\text{VT}},F\_{\theta})-\mathbf{D}\_{l}(\mathcal{D}\_{\text{T}},F\_{\theta})$ can be seen as an asymptotic Gaussian. To be specific, given a pooled sample standard deviation $\hat{\sigma}\_{l}=\sqrt{ \frac{ \sigma^{2}\_{l,{\text{T}}}} {|\mathcal{D}\_{\text{T}}|}+\frac{\sigma^{2}\_{l,{\text{VT}}}}{ |\mathcal{D}\_{\text{VT}}|} }$, we define:
>
> $\hat{\text{VIP}}(\mathcal{D},F\_\theta)=\arg\min\_{l\in \mathcal{L}\backslash L}\;\frac{\mathbf{D}\_{l}(\mathcal{D}\_{\text{VT}},F\_{\theta})-\mathbf{D}\_{l}(\mathcal{D}\_{\text{T}},F\_{\theta})}{\hat{\sigma}\_{l}}\geq\frac{\beta}{l-1}\sum\_{k<l}\frac{\mathbf{D}\_{k}(\mathcal{D}\_{\text{VT}},F\_{\theta})-\mathbf{D}\_{k}(\mathcal{D}\_{\text{T}},F\_{\theta})}{\hat{\sigma}\_{k}},$
>
> where $\sigma^{2}\_{l,\text{T}}=\frac{\sum\_{z\in\mathcal{D}\_{T}}(d(z^{l}\_{\text{vis}},z^{l}\_{\text{blind}})-\mathbf{D}\_{l}(\mathcal{D}\_{\text{T}},F\_{\theta}))^{2}}{|\mathcal{D}\_{\text{T}}|-1}$ is a sample variance on $\mathcal{D}\_{\text{T}}$ and $\sigma^{2}\_{l,\text{VT}}$ is similarly defined. Intuitively, the quantity on the left-hand side denotes a deviation-normalized representation distance in the layer $l$, whereas the right-hand side means the historical average of those distances with a weighting coefficient $\beta$, which can be set to 2.0 empirically. The above detection algorithm can be conducted on any given support dataset, and below we show its robust performance across different models and datasets.
>
> | Model (total number of layer) | VIP detection | MMBench (Spearman's $\rho$) | ViLP (Spearman's $\rho$) |
> | ------------- | -------- | ------- | ---- |
> | Qwen2.5-VL-7B (28) | Manual Selection |  18 (0.6335)      | 18 (0.6335)   |
> | Qwen2.5-VL-7B (28) | Variance-based Detection | 18 (0.6335)      | 19 (0.6336)  |
> | Gemma-3-4B (30)   | Manual Selection |    20 (0.7970)  |  20 (0.7970) |
> | Gemma-3-4B (30)  | Variance-based Detection | 16 (0.7973)    | 16 (0.7973)  |
> | InternVL3-8B (32)  | Manual Selection |   16 (0.5709)  |  16 (0.5709) |
> | InternVL3-8B (32)  | Variance-based Detection | 17 (0.5749)     | 20 (0.5949)  |
>
> We can see that VIP yielded by the automatic detection algorithms are quite close to the result of the manually selected one, and the Spearman correlations between TVIs computed by those estimated VIPs and the downstream prediction correctness are robust across different datasets. This result was also included in Appendix `Table 6` and `Figure 9` of the revised PDF file.

---

> ### Author Response · Authors · 2025-11-18
> **Response to Reviewer mW7M (2/3)**
>
> > _W2. More diverse fusion architectures._
>
> This is an excellent point. While our original model pool already includes several variations of multimodal fusion, we agree that many of them are indeed late-fusion-with-MLP-style. To address your concern, we have included additional analyses on **InstructBLIP-Vicuna-7B**, which adapts **Q-Former-style** fusion, in Appendix `Figure 11` in our revised draft. For **Flamingo-style** models, we have conducted experiments on **Llama-3.2-11B-Vision**, which incorporates multiple cross-attention layers for feature fusion, and have provided the results in `Figure 7`. Across these different architectures, we observe consistent VIP patterns and TVI trends. This strengthens our claim that VIP and TVI are not artifacts of specific architectural designs, but rather reflect a more general phenomenon in LVLMs.
>
> > _W3. It would also be valuable to test larger models (e.g., 70B-scale models)._
>
> Thanks for the suggestion. Due to the resource constraint, we limited the scope of our validation in our draft up to 27B-scale models. **We have now extended our experiments to the entire Qwen2.5-VL family**, which includes a 32B and 72B-scale model. Results provided in 11 show a similar trend that we have observed on Gemma-3 family. This further strengthens the generalizability of our conclusions.
>
> > _W4. An analysis of the failure patterns or outlier samples._
>
> We are really grateful for this insightful feedback! As you mentioned, although TVI shows a strong correlation with answer correctness, there are still some non-negligible portions of failure cases that TVI can not properly predict the model's answer quality.
>
> Conceptually, TVI is designed to capture the degree of effective visual integration during decoding. While higher TVI generally implies that the generated answer is more grounded in visual input and thus more likely to be correct, there are natural regimes where this relationship weakens, for example: (1) Some questions are not strongly visually demanding; the model can answer them correctly with relatively little visual integration. In such cases, the answer may be correct even when TVI is small, leading to "false negatives" from TVI's perspective. (2) For very challenging visual questions, the model may still fail even after substantial effort to integrate visual information (high TVI). This yields "false positives" in terms of TVI–correctness alignment. These factors introduce inherent noise into any evaluation based on TVI vs. correctness.
>
> To better understand such patterns, **we visualize the TVI distribution for ViLP samples that Qwen2.5-VL-7B answers correctly and perform a focused analysis on those with unexpectedly low TVI in** `Appendix E`. As shown in `Figure 17`, while the majority of correctly answered samples exhibit relatively high TVI, a small subset attains low TVI despite being answered correctly. Upon inspection, many of these cases can indeed be solved without relying on visual input, either due to knowledge leakage or lucky guessing. This suggests that such questions are intrinsically less visually demanding, and the low TVI observed in these cases does not constitute a failure of the metric but instead reflects the fundamental properties of the task itself.

---

> ### Author Response · Authors · 2025-11-18
> **Response to Reviewer mW7M (3/3)**
>
> > _W5. Including concrete examples of potential applications would make the work more actionable._
>
> We appreciate your concern. As our main goal of this study was to analyze the underlying behavior of models and provide diagnostic insights, we have consciously focused more on (1) devising an easy and simple analysis tool, (2) establishing a clear hypothesis, (3) collecting extensive evidence to validate the hypothesis, and (4) interpreting the results carefully.
>
> At the same time, we fully agree that presenting the concrete example to make use of our findings to improve LVLMs would provide more actionable takeaways. To this end, **we devised one possible application of our metric, using it as a regularization term** in addition to the cross-entropy loss for the next-token prediction task in vanilla visual instruction tuning. Specifically, we modified the LLaVA-v1.5-7B training objective to:
>
>
> $\mathcal{L}(\theta)=\mathbb{E}\_{X\sim \mathcal{D}}[-\log F\_{\theta}(X\_t | X\_{<t})-\lambda \cdot \text{TVI}(l^{*};X\_{<t},F\_{\theta})],$
>
> where we use a $\lambda$ (set to 0.03 by default) to control the strength of the regularization.
>
> Due to computational limitations, we finetuned on a 60k random subset of `llava_v1_5_mix665k`. All other hyperparameters used for visual instruction tuning remain default. The result is shown below as well as our revised manuscript `Table 5` (`Section 5`).
>
> |                        | MME (Perception) | MME (Reasoning) |
> | ---------------------- | ---------------- | --------------- |
> | Visual instruction tuning (vanilla)               | 1369.75          | 298.21          |
> | Visual instruction tuning w/ TVI regularization | **1400.44**          | **321.43**          |
>
> The performance improvement indicates that explicitly encouraging stronger visual integration (via TVI) leads to better downstream task performance. This result further shows the significance of TVI and suggests a potential practical use of TVI for the development of better LVLMs.
>
> > _W6. The evaluated models and benchmarks are not well-documented, even in the appendices. I recommend adding summary tables for both datasets and models._
>
> That's a great suggestion. By reflecting your comment, we add comprehensive documentation on models and datasets in `Table 8` (`Appendix D`), and below for your reference:
>
> | Model                  | Fusion      | # Layer  | Hidden Size    |
> | ---------------------- | ----------- | -------- | -------------- |
> | Qwen2.5-VL-7B/32B/72B  | MLP         | 28/64/80 | 3584/5120/8192 |
> | Gemma-3-4B/12B/27B     | Q-Former    | 34/48/62 | 2560/3840/5376 |
> | InternVL3-8B           | MLP         | 28       | 3584           |
> | LLaVA-v1.5-7B          | MLP         | 32       | 4096           |
> | Eagle2.5-8B            | MLP         | 28       | 3584           |
> | Llama-3.2-11B-Vision   | X-Attention | 40       | 4096           |
> | LLaVA-NeXT-Vicuna-7B   | MLP         | 32       | 4096           |
> | LLaVA-OV-Qwen2-7B      | MLP         | 28       | 3584           |
> | SmolVLM                | MLP         | 24       | 2048           |
> | InstructBLIP-Vicuna-7B | Q-Former    | 32       | 4096           |
>
> > _Q1. Do the authors observe any trend in VIP location or TVI magnitude with respect to model size? It may be difficult to compare across different architectures, but evaluating models of varying sizes within the same family in future work could help clarify this relationship._
>
> That is an important question, and we already provided the results across varying model sizes from `lines 397-418` in our manuscript with `Figure 5` as the result. We observed that the relative position of VIP (VIP layer / the number of total layer) is consistent across model scales, but the (dimension-normalized) average TVI increases along with the model size, indicating that larger and stronger models tend to leverage more visual information during answer generation.
>
> In addition, we performed further experiments on the Qwen2.5-VL family, which includes models up to 72B parameters. The same qualitative trends hold for these larger models as well. We have added a discussion of these results in `Figure 11` (`Appendix D`).

---

> ### Author Response · Authors · 2025-11-26
>
> Dear Reviewer mW7M:
>
> We hope you are doing well.
>
> We wanted to kindly remind you about our rebuttal. We have addressed all your insightful points, including _improving the clarity, adding additional experiments on models with diverse architectural designs and larger scales, exploring extra aggregation strategies for TVI, providing algorithm details for automatic VIP detection, adding further analysis on failure cases, and discussing the practical utility of TVI_. We sincerely hope our response offers satisfactory clarification.
>
> As the discussion period nears its close, we warmly invite any further questions you may have, and we look forward to the opportunity for continued discussion on these points.
>
> Thank you again for your time and thoughtful engagement!
>
> Sincerely,
> The Authors

---

### Official Review · Reviewer_PjBx · 2025-10-31

**Soundness:** 3
**Presentation:** 3
**Contribution:** 3
**Rating:** 6
**Confidence:** 4

**Summary:**

The authors introduce a chain-of-embedding (CoE) analysis that examines layer-wise hidden representations of LVLMs. By comparing embeddings produced with and without visual inputs across model layers, they identify a consistent Visual Integration Point (VIP) — the layer where visual information begins to meaningfully affect model behavior. Building on this, they define a quantitative measure called Total Visual Integration (TVI), which aggregates representation differences after the VIP to estimate how strongly the model integrates visual signals. A low TVI indicates a stronger language prior (i.e., text dominates), while a high TVI reflects effective multimodal grounding. The key findings are as follows: Every tested LVLM (9 models, 6 benchmarks) exhibits a distinct VIP layer where visual information starts influencing predictions. TVI strongly correlates with model correctness on visual reasoning tasks, outperforming traditional metrics like visual attention or output divergence. Theoretical analysis connects representation distance to KL divergence between visual and text-only distributions, offering an interpretable, information-theoretic view of LP..

**Strengths:**

1. The paper introduces chain-of-embedding analysis, a new internal probing method that captures layer-wise representation dynamics between visual and textual modalities — a perspective largely missing in prior language prior studies.
2. he discovery of the Visual Integration Point (VIP) provides an intuitive and mechanistic explanation of when LVLMs start to rely on vision, enhancing understanding beyond input–output probing.
3. The proposed Total Visual Integration metric offers a principled, continuous measure of multimodal reliance, showing strong correlation with model performance across tasks.
4. The authors test 9 LVLMs across 6 benchmarks (54 settings total), showing strong consistency and robustness of the findings.
Theoretical grounding: The work connects empirical findings to information-theoretic interpretations (via KL divergence), lending mathematical rigor to an otherwise empirical domain.

**Weaknesses:**

1. While the framework effectively analyzes where and how LP arises, it does not propose concrete strategies to reduce language prior or improve multimodal reasoning.
2. The method requires access to intermediate activations and model internals, making it impractical for closed-source LVLMs or large-scale commercial APIs.
3. While broad in scope, the work is heavily diagnostic, with limited discussion of causal factors or practical interventions to improve LVLMs.

**Questions:**

Questions are mentioned in the weakness

---

> ### Author Response · Authors · 2025-11-18
> **Response to Reviewer PjBx**
>
> We sincerely thank you for the positive comments and recognizing the strengths of our work. We also appreciate your valuable feedback on concrete applications and practical utility, which helps us further refine the manuscript. Below, we address your concerns in detail.
>
> > _W1 & W3. While the framework effectively analyzes where and how LP arises, it does not propose concrete strategies to reduce language prior or improve multimodal reasoning. While broad in scope, the work is heavily diagnostic, with limited discussion of causal factors or practical interventions to improve LVLMs._
>
> Thanks for this constructive concern. As our main goal of this study was to analyze the underlying behavior of models and provide diagnostic insights, we have consciously focused more on (1) devising an easy and simple analysis tool, (2) establishing a clear hypothesis, (3) collecting extensive evidence to validate the hypothesis, and (4) interpreting the results carefully.
>
> We believe such analytic and diagnostic work has intrinsic value, as it advances our understanding of modern neural networks through careful observation and characterization of their behavior. Examples in this spirit include double descent [1], neural collapse [2], Grokking [3], the Platonic representation hypothesis [4], as well as layer-wise representation dynamics [5].
>
> That said, we agree that demonstrating practical ways to leverage our findings would further strengthen the contribution. To that end, **we implemented one concrete application of our metric**: using TVI as an auxiliary regularization term, alongside cross-entropy loss for next-token prediction, during instruction tuning. Specifically, we modified the LLaVA-v1.5-7B training objective to:
>
>
> $\mathcal{L}(\theta)=\mathbb{E}\_{X\sim \mathcal{D}}[-\log F\_{\theta}(X\_t | X\_{<t})-\lambda \cdot \text{TVI}(l^{*};X\_{<t},F\_{\theta})],$
>
> where we use a $\lambda$ (set to 0.03 by default) to control the strength of the regularization.
>
> Due to computational limitations, we finetuned on a 60k random subset of llava_v1_5_mix665k. All other hyperparameters used for visual instruction tuning remain the default. The result is shown below as well as our revised manuscript `Table 5` (`Section 5`).
>
> |                        | MME (Perception) | MME (Reasoning) |
> | ---------------------- | ---------------- | --------------- |
> | Visual instruction tuning (vanilla)               | 1369.75          | 298.21          |
> | Visual instruction tuning w/ TVI regularization | **1400.44**          | **321.43**         |
>
> The performance improvement indicates that explicitly encouraging stronger visual integration (via TVI) leads to better downstream task performance. This result further shows the significance of TVI and suggests a potential practical use of TVI for the development of better LVLMs.
>
>
> > _W2. The method requires access to intermediate activations and model internals, making it impractical for closed-source LVLMs or large-scale commercial APIs._
>
> Thank you for highlighting this important limitation. We agree that our procedure relies on access to intermediate activations, a constraint we explicitly acknowledged in `Appendix H` (Limitations). This requirement is shared by most diagnostic and interpretability methods that analyze internal representations. Such access is essential because these internal states offer substantially richer and more reliable signals than output-only probing, enabling effective and mechanistic analysis of language priors.
>
> ---
>
> ### Reference
>
> [1] Nakkiran et al., "Deep Double Descent: Where Bigger Models and More Data Hurt," Journal of Statistical Mechanics: Theory and Experiment
>
> [2] Papyan et al., "Prevalence of neural collapse during the terminal phase of deep learning training," Proceedings of the National Academy of Sciences
>
> [3] Power et al., "Grokking: Generalization Beyond Overfitting on Small Algorithmic Datasets," arXiv:2201.02177
>
> [4] Huh et al., "The Platonic Representation Hypothesis," In ICML 2024
>
> [5] Skean et al., "Layer by Layer: Uncovering Hidden Representations in Language Models," In ICML 2025

---

> > ### Comment · Reviewer_PjBx · 2025-11-25
> >
> > Thank you for your response! I will keep my score positive.

---

> > > ### Author Response · Authors · 2025-11-26
> > >
> > > Thank you for your kind reply and for considering our response. Your feedback has been invaluable in improving our work!

---

### Official Review · Reviewer_2B1a · 2025-11-01

**Soundness:** 3
**Presentation:** 3
**Contribution:** 3
**Rating:** 6
**Confidence:** 3

**Summary:**

Vision-Language Models (VLMs) often rely overly on language priors. This paper examines this issue from a representation-level perspective by contrasting the internal embeddings of models with and without visual input. The authors propose two diagnostic metrics. The Visual Integration Point (VIP) identifies the layer where visual features begin to significantly influence textual representations. The Total Visual Integration (TVI) quantifies the overall magnitude of visual influence across layers. Experiments on multiple VLMs and benchmarks show that VIP positions are consistent within each model, while TVI correlates with both visual grounding strength and the dominance of language priors. The paper further analyzes the choice of distance metrics, the effect of model scale, and provides a theoretical justification.

**Strengths:**

1. The key idea, contrasting the internal embeddings with and without image input, is simple yet effective. It provides a direct way to study how visual information propagates within VLMs, offering a view of cross-modal interaction that complements existing output-based analyses.
2. The experiments produce consistent results across models and datasets, supporting the reliability of both proposed metrics. The findings are both insightful and well-evidenced through multiple analyses and visualizations.
3. The experiments cover a broad set of VLMs and benchmarks, and include examinations of distance metrics, model scale, and theoretical interpretation. This completeness strengthens the empirical grounding of the paper.

**Weaknesses:**

1. It states that "a higher TVI indicates that visual information exerts a stronger and more persistent influence throughout the later layers". However, the current TVI definition measures the mean visual effect after the integration point, which can overestimate models with brief but strong fusion peaks. Because it ignores the layer-wise distribution of visual influence, a single large spike can inflate TVI even if vision effects fade quickly. A refined version, such as a weighted or variance-penalized TVI, would better reflect sustained multimodal reasoning by emphasizing persistence and uniformity of visual influence.
2. Some statements, such as the claim that VIP corresponds to the onset of "effective problem-solving", are not fully supported by the evidence. Since the paper does not evaluate task performance at the layer level, VIP should be interpreted as marking representational change rather than functional reasoning. A more cautious phrasing is needed to improve accuracy and credibility.
3. The current analysis focuses primarily on the language-prior effect, contrasting text–image input with text-only input. A more complete triple-wise  comparison, including text-only, image-only and text-image pairs, would allow a deeper understanding of how each modality contributes individually and jointly to the representation flow, offering a fuller picture of multimodal integration.
4. While the work provides a valuable diagnostic for analyzing language priors, it stops short of offering guidance or methods to reduce such bias. Without follow-up analysis or strategies for mitigation, the practical utility of VIP and TVI remains limited to interpretation rather than its application.

**Questions:**

1. The paper introduces a threshold to determine the VIP but provides no clear method for how it is chosen or tuned for different models. It is stated that VIP is data independent but not model-dependent.
2. Although the anonymous code is provided, the shared link currently returns "The requested file is not found".
3. The use of the term "ablation" appears inaccurate. For example, the ablation on model scale is more of a sensitivity analysis, as it varies model size instead of removing the components. The authors might want to recap the research methodology.

---

> ### Author Response · Authors · 2025-11-18
> **Response to Reviewer 2B1a (1/3)**
>
> Thank you for recognizing the strengths of our work, particularly the simplicity and effectiveness of our method, as well as the strong empirical and theoretical evidence supporting it. We really appreciate the time you took to carefully review our paper and provide thoughtful, professional feedback. Your comments are very helpful for improving the manuscript. Below, we respond to each weakness (W) and question (Q) in turn.
>
> > _W1. Weighted or variance-penalized TVI._
>
> Thank you for this insightful suggestion. We have modified our phrasing in `lines 255-256` for better clarity. We also agree that considering layer-wise distribution structure can be beneficial for broader applicability and stability. To address this, we additionally implement a std–based reweighting scheme, which emphasizes layers whose distance distributions exhibit higher variability, i.e., layers that are more sample-specific and potentially more informative, while downweighting contributions that may arise from outliers or less discriminative layers:
>
> $\text{TVI}(l^\*;x,F\_{\theta})=\frac{1}{L-l^\* + 1}\sum\_{l=l^\*}^{L}[\sigma\_l\*d(z^{l}\_{\text{vis}},z^{l}\_{\text{blind}})],$
>
> where $\sigma_l$ denotes the standard deviation of representation distances at layer $l$. In `Table 9` (`Appendix E`) of our revised draft and below, we compare the effectiveness of this std-weighted TVI with that of the original (simple averaging) TVI on VLind-Bench and ViLP.
>
>
>
> | Model |  Aggregation             | VLind  | ViLP   |
> |--- | ------------- | ------ | ------ |
> | Qwen2.5-VL-7B | Simple Average | 0.7155 | 0.6335 |
> | Qwen2.5-VL-7B | Std Reweighting | 0.7164 | 0.6348 |
> | InternVL3-8B | Simple Average  | 0.6727 | 0.5709 |
> | InternVL3-8B | Std Reweighting  | 0.6739 | 0.5723 |
>
>
> The std-weighted TVI yields slightly better performance than the vanilla average, but the gain is modest and comes with additional computational overhead. These results suggest that our original, simpler TVI is already a strong and robust summary statistic, while the weighted variant can provide some refinement under specific scenarios that we now document in the paper.
>
> > W2. _VIP should be interpreted as marking representational change rather than functional reasoning._
>
> Thank you for raising this important point. To clarify our claim, we have provided an evaluation of task performance across layers, specifically, a comparison of pre-VIP and post-VIP performance in `Table 1`. This analysis shows a correlation between prediction correctness (task performance) and TVI before and after VIP, suggesting that VIP does mark a functional transition across layers to some extent.
>
> However, we agree that most of the evidence we have provided is more directly related to "representational change" rather than "functional change", and **we have relaxed our statement to avoid overclaim by reflecting your concern**. The change we made is highlighted in red in the revised PDF (See `line 295`).

---

> ### Author Response · Authors · 2025-11-18
> **Response to Reviewer 2B1a (2/3)**
>
> > _W3. Triple-wise comparison, including text-only, image-only and text-image pairs._
>
> Thank you for this suggestion. The last token hidden states of the text-only (question) and image-text paired (image + question) samples denote the model's answer-related representations. However, hidden states from image-only inputs----whether using the last token or the average of image-patch embeddings----**do not reflect the model’s answer prediction**, since no instruction is provided. As a result, these representations do not lie in the same behavioral space and are not directly comparable. Furthermore, most current LVLMs are not trained to handle image-only inputs for question answering. As a result, image-only representations may be out-of-distribution with respect to the tasks and analysis framework we use.
>
> For these reasons, and because our focus is specifically on LP during _response generation_, our main analysis focused on text-only and image-text conditions.
>
>
> Nevertheless, as you suggest, our contrasting chain-of-embedding framework is general and can be applied to other combinations of inputs. **We provide additional analyses contrasting image-text inputs and image-only inputs** in `Figure 14(a)` (`Appendix E`). In our experiments, the differences between the resulting curves are negligible. Despite that, we can see that the new TVI----which now captures how much the textual part of the input contributes to the decoding process----is slightly higher on vision-independent samples than on vision-dependent ones. This pattern aligns with expectation and further supports the robustness of our analysis framework and conclusions.
>
>
> Motivated by your suggestion, **we further explored the effect of instruction style** by comparing misleading instructions (which elicit LP) from original VLind-Bench and ViLP with vision-centric instructions such as "Describe the image in detail." As results in `Figure 14(b)` show, TVI under these vision-centric instructions is significantly higher than under misleading ones. This confirms that TVI can properly reflect the model's bias toward language priors under different instruction regimes.
>
> > _W4. Practical utility of VIP and TVI._
>
> We are glad that the diagnostic value of our analysis is appreciated! As our main goal of this study was to analyze the underlying behavior of models and provide diagnostic insights, we have consciously focused more on (1) devising an easy and simple analysis tool, (2) establishing a clear hypothesis, (3) collecting extensive evidence to validate the hypothesis, and (4) interpreting the results carefully.
>
> We fully agree that presenting the concrete example to make use of our findings to improve LVLMs would provide more actionable takeaways. To this end, **we devised one possible application of our metric, using it as a regularization term** in addition to cross entropy loss for next token prediction during instruction tuning. Specifically, we modified the LLaVA-v1.5-7B training objective to:
>
> $\mathcal{L}(\theta)=\mathbb{E}\_{X\sim \mathcal{D}}[-\log F\_{\theta}(X\_t | X\_{<t})-\lambda \cdot \text{TVI}(l^{*};X\_{<t},F\_{\theta})],$
>
> where we use a $\lambda$ (set to 0.03 by default) to control the strength of the regularization.
>
> Due to computational limitations, we finetuned on a 60k random subset of `llava_v1_5_mix665k`. All other hyperparameters used for visual instruction tuning remain the default. The result is shown below, as well as our revised manuscript `Table 5` (`Section 5`).
>
> |                        | MME (Perception) | MME (Reasoning) |
> | ---------------------- | ---------------- | --------------- |
> | Visual instruction tuning (vanilla)              | 1369.75          | 298.21          |
> | Visual instruction tuning w/ TVI | **1400.44**          | **321.43**          |
>
> The performance improvement indicates that explicitly encouraging stronger visual integration (via TVI) leads to better downstream task performance. This result further shows the significance of TVI and suggests a potential practical use of TVI for the development of better LVLMs.

---

> ### Author Response · Authors · 2025-11-18
> **Response to Reviewer 2B1a (3/3)**
>
> > _Q1. Threshold to determine the VIP._
>
> Thanks for pointing that out. We would like to clarify that VIP is **dataset-independent** while **model-dependent** (`lines 224-227`). VIPs used in our original manuscript are manually selected based on the observation on the empirical results of contrasting chain-of-embedding (ones like `Figure 3`). We provided an explanation of this process and the VIPs used for each model in `lines 1224-1230`, `Appendix C`.
>
> Besides, **we implemented an automatic VIP detection algorithm for potential large-scale application in practice**, by taking inspiration from our theoretical analysis (Lemma F.1) showing that our representation divergence term $\mathbf{D}\_{l}(\mathcal{D}\_{\text{VT}},F\_{\theta})-\mathbf{D}\_{l}(\mathcal{D}\_{\text{T}},F\_{\theta})$ can be seen as an asymptotic Gaussian. To be specific, given a pooled sample standard deviation $\hat{\sigma}\_{l}=\sqrt{ \frac{ \sigma^{2}\_{l,{\text{T}}}} {|\mathcal{D}\_{\text{T}}|}+\frac{\sigma^{2}\_{l,{\text{VT}}}}{ |\mathcal{D}\_{\text{VT}}|} }$, we define:
>
> $\hat{\text{VIP}}(\mathcal{D},F\_\theta)=\arg\min\_{l\in \mathcal{L}\backslash L}\;\frac{\mathbf{D}\_{l}(\mathcal{D}\_{\text{VT}},F\_{\theta})-\mathbf{D}\_{l}(\mathcal{D}\_{\text{T}},F\_{\theta})}{\hat{\sigma}\_{l}}\geq\frac{\beta}{l-1}\sum\_{k<l}\frac{\mathbf{D}\_{k}(\mathcal{D}\_{\text{VT}},F\_{\theta})-\mathbf{D}\_{k}(\mathcal{D}\_{\text{T}},F\_{\theta})}{\hat{\sigma}\_{k}},$
>
> where $\sigma^{2}\_{l,\text{T}}=\frac{\sum\_{z\in\mathcal{D}\_{T}}(d(z^{l}\_{\text{vis}},z^{l}\_{\text{blind}})-\mathbf{D}\_{l}(\mathcal{D}\_{\text{T}},F\_{\theta}))^{2}}{|\mathcal{D}\_{\text{T}}|-1}$ is a sample variance on $\mathcal{D}\_{\text{T}}$ and $\sigma^{2}\_{l,\text{VT}}$ is similarly defined. Intuitively, the quantity on the left-hand side denotes a deviation-normalized representation distance in the layer $l$, whereas the right-hand side means the historical average of those distances with a weighting coefficient $\beta$, which can be set to 2.0 empirically. The above detection algorithm can be conducted on any given support dataset, and below we show its robust performance across different models and datasets.
>
> | Model (total number of layer) | VIP detection | MMBench (Spearman's $\rho$) | ViLP (Spearman's $\rho$) |
> | ------------- | -------- | ------- | ---- |
> | Qwen2.5-VL-7B (28) | Manual Selection |  18 (0.6335)      | 18 (0.6335)   |
> | Qwen2.5-VL-7B (28) | Variance-based Detection | 18 (0.6335)      | 19 (0.6336)  |
> | Gemma-3-4B (30)   | Manual Selection |    20 (0.7970)  |  20 (0.7970) |
> | Gemma-3-4B (30)  | Variance-based Detection | 16 (0.7973)    | 16 (0.7973)  |
> | InternVL3-8B (32)  | Manual Selection |   16 (0.5709)  |  16 (0.5709) |
> | InternVL3-8B (32)  | Variance-based Detection | 17 (0.5749)     | 20 (0.5949)  |
>
>
> We can see that VIP yielded by the automatic detection algorithms are quite close to the result of the manually selected one, and the Spearman correlations between TVIs computed by those estimated VIPs and the downstream prediction correctness are robust across different datasets. This result was also included in Appendix `Table 6` and `Figure 9` of the revised PDF file.
>
>
>
> > _Q2. Although the anonymous code is provided, the shared link currently returns "The requested file is not found"._
>
> We are so sorry to hear that. At submission time, we verified that the anonymous code repository link was accessible. However, as noted in [the issue #548 in the official repository of anonymous GitHub](https://github.com/tdurieux/anonymous_github/issues/548), the hosting service temporarily experienced an outage during part of the reviewing period, which caused the link to fail intermittently. The issue has since been resolved, and the **same link is now working again**. The full code is currently accessible via the originally provided anonymous URL.
>
>
>
> > _Q3. The use of the term "ablation" appears inaccurate._
>
>
> We changed the paragraph name from "ablation on model scales" to "varying model scales" in the revised PDF (See `line 397` and the caption of `Figure 5`) by reflecting your recommendation.

---

> ### Author Response · Authors · 2025-11-26
>
> Dear Reviewer 2B1a:
>
> We hope you are doing well.
>
> We wanted to kindly remind you about our rebuttal. We have addressed all your insightful points, including _improving the clarity, adding additional experiments on triple-wise comparison, exploring extra aggregation strategies for TVI, providing algorithm details for automatic VIP detection, and discussing the practical utility of TVI_. We sincerely hope our response offers satisfactory clarification.
>
> As the discussion period nears its close, we warmly invite any further questions you may have, and we look forward to the opportunity for continued discussion on these points.
>
> Thank you again for your time and thoughtful engagement!
>
> Sincerely,
> The Authors

---

> ### Comment · Reviewer_2B1a · 2025-11-27
>
> Thank you for the detailed response. I enjoyed it and it does address most of my concerns. Given your commitment to the proposed revisions and my updated understanding of the paper, I am willing to raise my score.
> (It seems like now I can not edit my original review. It looks like an openreview system issue.)

---

> > ### Author Response · Authors · 2025-11-28
> >
> > Thank you very much for your positive feedback and your willingness to raise the score. We are pleased to hear that our response has addressed your concerns!
> >
> > We understand your difficulty in editing the score. This is likely an OpenReview system issue. We kindly ask you to please follow up and update your score when the system permits.
> >
> > Thank you again for your support.

---

### Official Review · Reviewer_zCDF · 2025-11-04

**Soundness:** 3
**Presentation:** 2
**Contribution:** 2
**Rating:** 4
**Confidence:** 4

**Summary:**

This paper introduces a framework to analyze the internal mechanisms of MLLMs, specifically their over-reliance on language priors instead of visual evidence. The authors propose a "chain-of-embedding" analysis, which contrasts layer-wise hidden states from vision-text inputs with those from text-only inputs. This analysis reveals two key findings. First, all models exhibit a Visual Integration Point, a specific layer where visual information begins to meaningfully influence representations. Second, the paper introduces the Total Visual Integration estimator, which aggregates representation distances after the VIP to quantify the strength of visual integration per sample. A high TVI indicates effective visual grounding, while a low TVI suggests a strong LP reliance. This framework was consistently validated across 54 model-dataset combinations.

**Strengths:**

1. Novel Analysis of Language Prior. The paper proposes a novel framework for analyzing Language Prior in LVLMs. Instead of relying on traditional input-output probing , it introduces a fine-grained, white-box methodology that examines the internal, layer-wise "chain-of-embedding". This provides a new and insightful perspective on how and when models integrate visual information.
2. Sufficient Experimental Validation. The authors demonstrate the robustness of their claims through extensive experiments. The existence of the Visual Integration Point (VIP) is not an isolated finding; it is consistently observed across 9 different LVLMs and 6 distinct benchmarks, totaling 54 experimental settings. This large-scale validation strongly supports the claim that the VIP is a universal phenomenon in these architectures.

**Weaknesses:**

1. Lack of Clarity in Methodology. The main body of the paper omits critical methodological details, hindering clarity and reproducibility.
	- Undefined Symbols in Formulas. In Section 3.1, Equation (2) introduces the notations $P_{star}$ and $D_{star}$ without explicitly defining what the wildcard $star$ represents. While context implies it refers to the $P_{VT}$ and $P_T$ distributions defined earlier, this omission makes the formulation imprecise.
	- Ambiguous Distance Calculation. The paper's core distance metric, $d(z_{vis}^l, z_{blind}^l)$, is underspecified in the main text. Both $Z^l_{vis}$ and $Z^l_{blind}$ are sequences of token embeddings, not single vectors. The crucial methodological detail—that the distance is only computed using the hidden state of the last token—is only mentioned in the appendix. This choice is non-trivial, as it impacts the interpretation of the results, and should be explicitly defined and justified in Section 3.1.
2. Lack of Interventional Validation for TVI. The paper's validation for the Total Visual Integration metric is purely diagnostic. The authors demonstrate that TVI correlates strongly with task correctness on vision-dependent benchmarks . However, this is not sufficient proof of its reliability as a proxy for Language Prior. A much stronger validation would be an interventional experiment. The authors mention existing methods for mitigating LP (e.g., contrastive decoding, attention reallocation) in their related work . They should have applied these mitigation techniques and then measured the TVI score. If TVI is a reliable metric, it should significantly increase (reflecting lower LP) when these mitigation methods are active.

**Questions:**

1. Can this solution be used for data filtering to screen for sample pairs that are strongly visually relevant?

2. Looking at Figures 2 and 3, the difference in representation distance after VIP is only on the order of 0.001. Is this difference not significant enough?

3. Why only use the last token hidden states for calculation? Can all embeddings be included?

4. Is the VIP layer different at different model training stages? What would the trend be?

---

> ### Author Response · Authors · 2025-11-18
> **Response to Reviewer zCDF (1/2)**
>
> We sincerely appreciate your recognition of the novelty of our analysis on language prior and the comprehensiveness of our experimental validation. Moreover, thank you so much for taking the time to review our work rigorously and providing valuable feedback on methodological clarity and evidential completeness. Your comments have greatly helped us improve the manuscript. Below, we respond to each weakness (W) and question (Q).
>
> > _W1. Lack of Clarity in Methodology: Undefined Symbols in Formulas; Ambiguous Distance Calculation._
> > _Q3. Why only use the last token hidden states for calculation? Can all embeddings be included?_
>
> Thank you for raising these concerns. Following your suggestion, we have revised the manuscript by: (1) adding an explicit definition of $D_{\star}$ and $P_{\star}$ in `line 158` of `Section 3.1` rather than leaving them inferred from the context; (2) explicitly specifying the embedding $Z^{l}$ as the last token embedding in `lines 149-150`, along with a justification in `Section 3.1`. We have also included a link to our code to ensure full reproducibility.
>
>
> Our choice to use the last-token embedding aligns with standard practice in prior work on LLM self-evaluation and hallucination detection [1,2,3]. In autoregressive models, the last-token hidden state **encodes information about all preceding tokens**, since each position attends to all earlier positions via causal self-attention. Here, the token embedding in the answer position (the last token) aggregates all the previous information in order to produce a valid response.
>
> That said, we also incorporated your suggestion to compare last-token TVI with all-token-averaged TVI. Our results show that averaging over all tokens introduces substantial noise and noticeably weakens the predictive reliability of TVI.
>
> |                              | ViLP   | VLind  |
> | ---------------------------- | ------ | ------ |
> | Last-token TVI             | **0.6335** | **0.7155** |
> | All-token-averaged TVI | 0.4906 | 0.5592 |
>
>
> > _W2. Interventional validation for TVI._
>
> We fully agree that testing TVI under mitigation interventions would help to rigorously demonstrate the reliability of TVI as an indicator of LP. Following your suggestion, we applied a popular hallucination mitigation approach PAI [4] to Qwen2.5-VL-7B. This intervention directly modifies visual attention maps and intervenes the hidden representations for analyzing TVI dynamics. Below we report TVI and task accuracy on ViLP before and after applying the intervention. We have also included the full results in `Table 4` (`Section 4`) in the revised manuscript.
>
> |  | Accuracy (%)| TVI |
> | ------------------- | ------ | ---- |
> | Before intervention | 50.00 | 0.038 |
> | After intervention [4]  | **52.33** | **0.144**|
>
>
> As we can see, **interventions that improve performance also lead to a notable increase in TVI**. This result indicates that TVI genuinely captures the deep effect of visual integration, and thus is indeed a reliable metric that robustly quantifies LP.
>
> To further assess the causal role of TVI, we also experimented with explicitly optimizing TVI via a regularization term during next-token-prediction-based visual instruction tuning. Specifically, we modified the LLaVA-v1.5-7B training objective to:
>
> $\mathcal{L}(\theta)=\mathbb{E}\_{X\sim \mathcal{D}}[-\log F\_{\theta}(X\_t | X\_{<t})-\lambda \cdot \text{TVI}(l^{*};X\_{<t},F\_{\theta})],$
>
> where we use a $\lambda$ to control the strength of the regularization.
>
> Due to computational limitations, we finetuned on a 60k random subset of `llava_v1_5_mix665k`. All other hyperparameters used for visual instruction tuning remain the default. The result is shown below, and has been added to our manuscript in `Table 5` (`Section 5`).
>
> |                        | MME (Perception) | MME (Reasoning) |
> | ---------------------- | ---------------- | --------------- |
> | Visual instruction tuning (vanilla)              | 1369.75          | 298.21          |
> | Visual instruction tuning w/ TVI | **1400.44**          | **321.43**          |
>
> The performance improvement indicates that **explicitly encouraging stronger visual integration (via TVI) leads to better downstream task performance**. This not only supports TVI as a reliable metric but also highlights its potential as a practical tool for improving LVLMs.
>
> ---
>
> ### Reference
>
> [1] Jiang et al., "On Large Language Models’ Hallucination with Regard to Known Facts," In NACCL 2024
>
> [2] Tian et al., "Toward Self-Improvement of LLMs via Imagination, Searching, and Criticizing," In NeurIPS 2024
>
> [3] Li et al., "CoRE: Enhancing Metacognition with Label-free Self-evaluation in LRMs," arXiv:2507.06087
>
> [4] Liu et al., "Paying More Attention to Image: A Training-Free Method for Alleviating Hallucination in LVLMs," In ECCV 2024

---

> ### Author Response · Authors · 2025-11-18
> **Response to Reviewer zCDF (2/2)**
>
> > _Q1. Can this solution be used for data filtering to screen for sample pairs that are strongly visually relevant?_
>
> That's a very interesting question. In practice, it is hard to directly adopt our TVI as a data filtering solution. This is because TVI only captures the strength of effective visual integration and language prior. A low TVI can arise from: (1) intrinsically low vision-dependence of the sample, (2) misleading or biased linguistic patterns in the data, and (3) the model’s own robustness (or fragility) to language priors, etc.
>
> Since these factors are entangled, TVI alone cannot reliably serve as a data-filtering criterion without additional assumptions. However, if we have external control or prior knowledge about the model's LP, TVI becomes much more interpretable and can be used for identifying visually dependent samples.
>
>
>
> > _Q2. Looking at Figures 2 and 3, the difference in representation distance after VIP is only on the order of 0.001. Is this difference not significant enough?_
>
> Thanks for pointing out this important point. As each model has different hyperparameter configurations (such as the size of hidden dimensions) and learned embedding structures, **the cosine distance scale can vary significantly depending on the model family and architecture**. Therefore, the magnitude (e.g., 0.001) is not directly meaningful on its own. What matters is the relative difference between the representation distances on $\mathcal{D}\_\text{VT}$ vs. $\mathcal{D}\_\text{T}$ at pre- and post-VIP layers.
>
> To this end, in all our line plots, we visualized a standard deviation of representation distances as the shadow region to stress how significant the difference is. **The significance should be assessed through the relative difference given the standard deviation rather than the absolute scale**. As shown in `Figures 2, 3`, and `5` (with standard deviation shading), the pre- vs. post-VIP divergence is highly consistent and statistically significant, even if the absolute values are small.
>
> Besides, `Figure 5` includes dimension-normalized results, enabling scale-invariant comparison across models.
>
>
>
> > _Q4. Is the VIP layer different at different model training stages? What would the trend be?_
>
> That's a good point! Exploring how VIP evolves across different stages of model training is definitely interesting. To address this, we evaluated checkpoints from multiple stages of LLaVA-v1.5 visual instruction tuning. The resulting plot is included in `Figure 15` and `Figure 16` (`Appendix E`).
>
>
> The key findings are: (1) **The VIP position remains remarkably stable across training stages (e.g., consistently at 12 for LLaVA-1.5-7B; see `Figure 15`).** This suggests that VIP appears to reflect an intrinsic architectural property rather than a behavior shaped by instruction tuning. (2) **TVI exhibits a clear upward trend as training progresses**, indicating that the degree to which the model effectively incorporates visual information improves gradually during instruction tuning (`Figure 16`). These findings align with our previous conclusion that VIP is a model-specific property as well as our expectation that visual instruction tuning incrementally enhances the model’s multimodal fusion capability.

---

> ### Author Response · Authors · 2025-11-26
>
> Dear Reviewer zCDF:
>
> We hope you are doing well.
>
> We wanted to kindly remind you about our rebuttal. We have addressed all your insightful points, including _improving the clarity, adding additional experiments on interventional validation, exploring extra aggregation strategies for TVI, and discussing the practical utility of TVI_. We sincerely hope our response offers satisfactory clarification.
>
> As the discussion period nears its close, we warmly invite any further questions you may have, and we look forward to the opportunity for continued discussion on these points.
>
> Thank you again for your time and thoughtful engagement!
>
> Sincerely,
> The Authors

---

### Author Response · Authors · 2025-11-18
**General Response**

Dear Reviewers, AC, SAC, and PC,

We sincerely appreciate the time and effort you have devoted to reviewing our manuscript.

In this paper, we propose a formal framework to understand and quantify the language prior of LVLMs through the lens of their internal states. We are grateful for the reviewers' recognition of the novelty and soundness of our approach, as well as the comprehensiveness and rigor of our experimental validation. For convenience, we summarize below the key strengths highlighted by the reviewers, followed by the corresponding revisions we have made in response to the concerns:

---

### Key strengths noted by the reviewers:

- S1: _Novel and insightful representation-level analysis framework_ by contrasting chain-of-embedding, offering a new perspective on language priors. (All reviewers)
- S2: _Simple yet effective metrics_ that are easy to implement and _theoretically grounded_. (R2, R3, R4)
- S3: _Extensive and rigorous experimental validation_ across 9 LVLMs and 6 benchmarks (54 settings). (All reviewers)
- S4: _Goes beyond traditional input–output probing_ and fills a key methodological gap in LP research by analyzing internal layer-wise dynamics. (R1, R3, R4)
- S5: Provides _intuitive mechanistic insights_ into when and how vision influences LVLM behavior (e.g., discovery of VIP). (All reviewers)
- S6 _Strong correlation of TVI with visually grounded correctness_, outperforming prior LP-related metrics. (All reviewers)

---

### Key revisions in response to the comments:


We have carefully revised the manuscript, and the main updates are summarized below:

- [R1, R2, R3, R4]: Added **further discussion on the pratical utility of TVI** and provided a concrete example demonstrating how TVI can be used to guide the development of better LVLMs (`Section 5`).
- [R2, R4]: Specified the **algorithmic details for automatic VIP detection** and provided corresponding experimental results (`Appendix B`).
- [R4]: Extended experiments to **models with more diverse architectures and larger scales** (`Appendix D`).
- [R1, R2, R4]: Added **additional analyses** to further deepen the understanding of our framework, including different aggregation strategies for TVI computation, instruction-level perturbation, image–text vs. image-only chain comparison, VIP and TVI evolution across training stages, interventional validation of TVI, and case studies on TVI failure cases (`Section 4` & `Appendix E`).
- [R1, R2]: Improved phrasing, clarified definitions and motivations, and addressed several presentation issues (`Section 3 & 4`).

All the revised content is highlighted in red.

(* We refer to Reviewer zCDF as R1, Reviewer 2B1a as R2, Reviewer PjBx as R3, and Reviewer mW7M as R4)

---

### Author Response · Authors · 2025-12-02
**Post-rebuttal Summary**

Dear AC, SAC, and PC,

We sincerely appreciate the time and effort you are dedicating to reaching an objective decision on our submission, especially under the unusual circumstances of the current ICLR cycle.

This paper investigates a **fundamentally important yet underexplored** problem for modern LVLM development:
> How do LVLMs internally integrate visual information vs. relying on language priors?

Existing research on language prior is extremely sparse, lacking empirical effectiveness or failing to provide insights into the model's internal modality integration. Our work addresses the crucial gap with a novel approach that, for the first time, identifies _where and how_ visual signals  influence LVLMs' internal representations through the discovery of _Visual Integration Point_ (VIP). We further propose a new metric, Total Visual Integration (TVI), that effectively quantifies language prior strength. These proposals are grounded by extensive empirical validation across 9 architectures and 6 datasets (54 total settings), providing a practical and reliable diagnostic for language priors, and offering actionable and theoretical insight for designing more capable LVLMs in the future.


Across the reviews, there was **unanimous recognition of the work’s novelty, analytical rigor, solid experiments, and value to the community**. We summarize the post-rebuttal reviewer stance below:

---

**Post-rebuttal stance**

- Reviewer 2B1a: Maintained a positive and constructive position and explicitly **strengthened the stance for acceptance**, and indicated intent to raise score from **6 → 8**:

    > "_I enjoyed it and it does address most of my concerns. Given your commitment to the proposed revisions and my updated understanding of the paper, I am willing to raise my score_."

- Reviewer PjBx has been **supportive for acceptance** from the beginning and reaffirmed the positive position after discussion (**score 6**):
    > "_Thank you for your response! I will keep my score positive._"

- Reviewer zCDF and Reviewer mW7M both positively endorsed (1) the novelty of our approach, (2) theoretical grounding, and (3) extensiveness of our empirical validation:
    > "_The paper proposes a novel framework for analyzing Language Prior in LVLMs...This provides a new and insightful perspective on how and when models integrate visual information._"

    > "_The newly proposed metric, TVI, offers a more robust and empirically and theoretically grounded way to quantify visual grounding_."

    > "_The authors demonstrate the robustness of their claims through extensive experiments...This large-scale validation strongly supports the claim that the VIP is a universal phenomenon in these architectures._"

    Their initial rating of 4 was primarily due to concerns about notation clarity and generalizability of our findings to broader architectures and larger models---rather than the methodology itself. We addressed these concerns by (i) fixing the notation definition and clarifying the use of last-token embedding, (ii) adding interventional analyses for TVI, (iii) extending experiments to include Q-Former and cross-attention fusion models, as well as larger models of up to 72B-scale. In addition, we added further discussion on the practical utility of TVI and provided a concrete example demonstrating how TVI can be used to guide the development of better LVLMs.

    Due to ICLR’s communication freeze after November 28, we were unfortunately unable to receive further follow-up; however, **all concerns raised by these two reviewers were fully addressed point-by-point in our rebuttal and revised manuscript, supported by new evidence**.

In closing, the revisions made during rebuttal significantly strengthened the rigor, clarity, and practical relevance of the work. We have also provided a summary of the manuscript updates [here](https://openreview.net/forum?id=wnTnwWKu72&noteId=kiNjwZ8m4O). We believe the work provides a timely and important contribution to the ICLR community, and we are grateful to the reviewers and AC for their careful engagement.

Sincerely,

The Authors

---

### Meta-Review · Area_Chair_13Hb · 2026-01-09

**Summary:**

The reviewers broadly agree that the paper presents a novel and technically sound analysis framework for understanding language prior in LVLMs by examining internal representations rather than relying on input–output probing. The introduction of the Visual Integration Point and the Total Visual Integration metric is generally viewed as insightful, well motivated, and supported by a large-scale empirical study across many models and benchmarks. The main concerns that initially limited enthusiasm were not about correctness, but about clarity, interpretation, and scope: several reviewers questioned whether some claims around VIP were overstated, whether the TVI aggregation could misrepresent transient fusion effects, and whether the work was too purely diagnostic without clear guidance on how the proposed tools could be used to improve models. Additional concerns included incomplete methodological specification in the original draft, lack of interventional validation, architectural diversity, and evaluation on larger models. These issues placed the paper in a borderline-to-positive range prior to rebuttal, with some reviewers explicitly unsure whether the contribution cleared the acceptance bar.

**Reviewer Concerns:**

The rebuttal was unusually thorough and addressed essentially all substantive technical concerns. The authors clarified notation and methodological choices, added interventional validation demonstrating that TVI responds meaningfully to mitigation strategies, extended experiments to more diverse fusion architectures and larger models (up to 72B scale), and provided additional analyses on aggregation strategies, failure cases, and VIP stability across training stages. Several reviewers’ concerns about overclaiming were handled by tightening language and adding supporting evidence. Importantly, the authors also went beyond a purely diagnostic framing by demonstrating a concrete application of TVI as a regularizer during instruction tuning, which directly addressed concerns about practical utility. At this point, the remaining limitations are largely intrinsic to the scope of the work rather than unaddressed flaws: the method requires access to internal activations and remains primarily an analysis tool, and the proposed application is only one example rather than a full mitigation framework. These are reasonable constraints and are clearly acknowledged in the revised manuscript.

**Reviewer Scores:**

Reviewer zCDF initially scored the paper below the acceptance threshold, with concerns focused on missing methodological details, lack of interventional validation, and ambiguity in interpreting VIP and TVI. These concerns were valid and substantive. The authors addressed them point by point with added experiments, clearer definitions, and new analyses. However, zCDF did not participate further in the discussion after rebuttal. In the absence of explicit follow-up, the AC believes the most conservative interpretation is that the rebuttal would significantly reduce the reviewer’s reservations, but it is unclear whether it would have led to a definite score increase. Reviewer 2B1a initially gave a marginally positive score and raised thoughtful concerns about the interpretation of TVI, overclaiming around VIP, and practical usefulness. This reviewer actively participated in the discussion and explicitly stated that the rebuttal addressed most concerns and that they were willing to raise their score, indicating a clear upward revision. Reviewer PjBx was positive from the outset, raised concerns mainly about the diagnostic nature of the work and applicability to closed models, and participated in the discussion to confirm that their score would remain positive; there is no indication their score would change. Reviewer mW7M initially scored the paper below the threshold, citing architectural diversity, scalability, and sensitivity of VIP detection. These concerns were directly addressed with additional experiments and analyses, but the reviewer did not participate further in discussion. As with zCDF, the AC views the rebuttal as substantially strengthening the paper, though there is no explicit signal of a score update.

---

### Decision · Program_Chairs · 2026-01-26

Accept (Poster)